

# Microphysics of Summer Clouds in Central West Antarctica Simulated by Polar WRF and AMPS

5  Keith M. Hines[1], David H. Bromwich[1,2], Sheng-Hung Wang[1], Israel Silber[3], Johannes Verlinde[3], Dan Lubin[4]

[1]Polar Meteorology Group, Byrd Polar & Climate Research Center, The Ohio State University, Columbus, OH, 43210, USA

[2]Atmospheric Sciences Program, Department of Geography, The Ohio State University, Columbus, OH, 43210, USA

[3]Department of Meteorology and Atmospheric Sciences, The Pennsylvania State
15  University, University Park, PA, 16802, USA

[4]Scripps Institution of Oceanography, University of California, San Diego, La Jolla, CA, 96802, USA

Submitted to Atmospheric Chemistry and Physics

25                              November 2018

*Correspondence to*: Keith M. Hines (hines@polarmet1.mps.ohio-state.edu)





**Abstract.** The Atmospheric Radiation Measurement (ARM) West Antarctic Radiation Experiment (AWARE) provided a highly detailed set of remote sensing and surface observations to study Antarctic clouds and surface energy balance, which have received much less attention than for the Arctic due to greater logistical challenges. Limited prior Antarctic cloud observations has slowed the progress of numerical weather prediction in this region. The AWARE observations from WAIS Divide during December 2015 and January 2016 are used to evaluate the operational forecasts of the Antarctic Mesoscale Prediction System (AMPS) and new simulations with Polar WRF 3.9.1. The Polar WRF 3.9.1 simulations are conducted with advanced microphysics schemes and with the WRF single-moment 5-class microphysics (WSM5C) also used by AMPS. AMPS simulates few liquid clouds during summer at WAIS Divide, inconsistent with observations of frequent low-level liquid clouds. Polar WRF 3.9.1 simulations show that this result is a consequence of WSM5C. More advanced microphysics schemes simulate more cloud liquid water and produce stronger cloud radiative forcing, resulting in downward longwave and shortwave radiation at the surface more in agreement with observations. Similarly, increased cloud fraction is simulated with the more advanced microphysics schemes. All of the simulations, however, produce smaller net cloud fractions than observed. Ice water paths vary less between the simulations than liquid water paths. The colder and drier atmosphere driven by GFS initial and boundary conditions for AMPS forecasts produces lesser cloud amounts than the Polar WRF 3.9.1 simulations driven by ERA-Interim.

## 1 Introduction

West Antarctica is among the most rapidly warming locations on Earth, and its warming is closely linked with global sea level rise (Rignot, 2008: Turner et al., 2006; Steig et al., 2009; Bromwich et al., 2013a, 2014). Recent paleoclimate work links temperature increases of a few degrees with past sea level increases of several meters due to disintegration of parts of the Antarctic Ice Sheet (DeConto and Pollard, 2016). Additional rise in Antarctic summer temperatures could lead to more frequent and extensive surface melting of the West Antarctic Ice Sheet (WAIS) (e.g., Nicolas and Bromwich, 2014). Conversely, increased temperatures can result in greater evaporation over the oceans and increased snowfall over Antarctica (Nicolas and Bromwich, 2014). The observational evidence shows West Antarctic warming since the 1950s (Bromwich et al., 2013a). Yet, there is disagreement about the cause, magnitude, seasonality and spatial extent of this warming due to regional and temporal gaps in the observational record and the overlapping influences of the El Niño-Southern Oscillation, the Southern Annular Mode, greenhouse gases, and ozone (e.g., Fogt et al., 2011; Bromwich et al., 2013a; Clem and Fogt, 2013; Hosking et al., 2016; Nicolas et al., 2017; Screen and Simmonds, 2012).

Unlike the elevated ice mass of East Antarctica, West Antarctica is highly prone to intrusions of moist air from the Southern Ocean (Nicolas and Bromwich, 2011, Scott et al., 2017). Thus, the West Antarctic climate is much more ocean-dominated





than that of the colder and drier East Antarctica. Moisture flux over West Antarctica leads to cloud formation. Clouds alter the net surface radiative flux and can thus impact the onset, extent, intensity, and duration of surface melting, refreezing, and ultimately meltwater control on cryospheric dynamics or runoff into the ocean (van Trinch et al., 2016). Modelling studies have shown that changes in cloud properties over Antarctica may impact regions of the globe well beyond high southern

latitudes (Lubin et al., 1998). Moreover, Antarctic clouds have different characteristics than Arctic clouds (Hogan 1986; Bromwich et al., 2012; Grosvenor et al., 2012; O'Shea et al. 2017). Antarctica can have very low ice nuclei (IN) concentrations (Hogan, 1986; Grosvenor et al., 2012; O'Shea et al., 2017). Silber et al. (2018a) show that cloud thickness at McMurdo Station peaks in austral winter, possibly due to cyclone activity, while Arctic cloud thickness peaks in boreal summer. O'Shea et al. (2017) note significantly different types and concentrations of cloud condensation nuclei and IN are expected between the

Arctic and Antarctic due to the minimal anthropogenic sources at high southern latitudes. Consequently, it's uncertain how well the findings of the various Arctic field programs and modelling experiments translate to Antarctica.

Clouds, including liquid water clouds, have a strong modulation on the local climate (Nicolas and Bromwich, 2011; Bromwich et al., 2012; Scott et al., 2017; Silber et al., 2018a). A supercooled liquid cloud is likely to be more optically thick than a fully

glaciated ice cloud (Shupe and Intrieri, 2004; Grosvenor et al., 2012; McCoy et al., 2015). Unfortunately, there have been few Antarctic field programs to detail cloud microphysical properties (e.g., Bromwich et al., 2012; Lachlan-Cope et al., 2016; Scott and Lubin, 2016). One study in the past decade by the British Antarctic Survey examined clouds over the Antarctic Peninsula (e.g., Grosvenor et al., 2012; Lachlan-Cope et al., 2016). Lachlan-Cope et al. (2016) found large differences in ice crystal concentrations between the clouds on the eastern and western sides of the peninsula, while Grosvenor et al. (2012) found

elevated ice crystal concentrations with relatively warm temperature between -0.4 and -6.6°C. They also found that several widely used IN parameterizations poorly represented the observed relationship between ice particle concentration and temperature. Accordingly, clouds are frequently poorly represented in numerical simulations for Antarctica (e.g., Bromwich et al., 2013b; King et al., 2015). The following sections discuss efforts to evaluate and improve the simulation of Antarctic clouds. The recent AWARE project is discussed in Sect. 2, while Sect. 3 describes the Polar WRF simulations for this project,

including AMPS numerical weather prediction forecasts for Antarctica. Results are discussed in Sect. 4, and Conclusions are given in Sect. 5.

## 2 AWARE

The Atmospheric Radiation Measurement (ARM) West Antarctic Radiation Experiment (AWARE, Witze et al. 2016) is a recent robust field program to study clouds and their impacts on atmospheric radiative transfer over the Antarctic continent.

The prime motivation for AWARE was that there have been no substantial atmospheric science or climatological field work on WAIS in decades using suites of advanced equipment (Witze et al., 2016) other than a few automatic weather stations that provided direct meteorological information since 1980 (Lazzara et al., 2012).



AWARE used the joint capabilities of the U.S. Antarctic Program, managed by the National Science Foundation, and the Department of Energy's second ARM Mobile Facility (AMF2) to provide quantitative data about energy components, changing air masses, and cloud microphysical data to improve model simulations of the ice sheet as influenced by earth system processes. The AMF2 consists of a collection of lidars, radars, and radiometers taking remote-sensing observations of the

Antarctic clouds combined with in situ instruments documenting the atmospheric state, but more comprehensive observations are needed. There is a need to quantify the impact of continental and oceanic air masses on the local hydrology and surface energy balance. Furthermore, there is a need for observations that can enable improved numerical simulations, both regional and global, through better representation of Antarctic clouds. The scarcity of cloud observations and well-tested simulations has so far inhibited significant progress.

Beginning late November 2015, AMF2 was deployed to Antarctica to make the first well-calibrated climatological suite of measurements in more than 40 years (Witze et al., 2016). The primary AWARE site was McMurdo Station (77.85°S, 166.72°E) at the southern tip of Antarctica's Ross Island where observations took place between November 2015 and January 2017. A smaller suite of instruments was also deployed to WAIS Divide (79.468°S, 112.086°W, 1803 m above sea level) for 56 days

during the early and middle parts of austral summer (November 2015 - January 2016).

The WAIS Divide component of the AWARE field campaign ran from 4 December 2015 through 18 January 2016. A suite of ARM Mobile Facility instruments (Mather and Voyles, 2013) optimized for surface energy budget observations was moved from McMurdo to the WAIS Divide site during this period. Estimates of upper-air temperature and moisture were obtained

from six-hourly rawinsonde launches and continuous retrievals from a profiling microwave radiometer (MWR, Morris 2006). Liquid water path (LWP) was extracted from a co-location of the MWR with a G-Band Vapor Radiometer Profiler (Cadeddu, 2010).

Upwelling shortwave and longwave radiative flux components were measured by a Surface Energy Balance system (SEBS,

Cook, 2018). Downwelling flux components were measured by a Sky Radiation System, which consists of a normal incidence pyrheliometer, shaded pyranometers and pyrgeometers (Dooraghi et al., 1996). The global downwelling shortwave flux was computed as in Nicolas et al. (2017). Surface fluxes for sensible and latent heat are derived according to the algorithm of Andreas et al. (2010). Near-surface measurements of temperature, moisture and wind speed were measured by the ARM surface meteorological instrumentation (Holdridge and Kyrouac, 1993). Furthermore, estimates of the conductive heat flux

from the ice surface and the underlying ice were taken from Nicolas et al. (2017) who calculated the residual of other terms in the surface energy balance.

A cloud mask (derived from detected hydrometeor-bearing air-volumes) is used to determine the cloud and liquid occurrence fractions at WAIS Divide associated with the method of Silber et al. (2018a). In brief, depolarization micropulse lidar (MPL;



Flynn et al., 2007) observations which were processed at Penn State (Silber et al., 2018b) are used to generate a linear depolarization ratio (LDR) versus log-scaled particulate backscatter cross-section two-dimensional histogram. This histogram, which is based on the full MPL data set from the WAIS Divide (after rough estimation of the particulate backscatter cross-section; see Silber et al., 2018c), is utilized for the determination of the hydrometeor population boundaries within the LDR-

backscatter parameter space, followed by the classification of the lidar returns.

Hourly time series of total hydrometeor and liquid-cloud fractions were calculated from the processed cloud and liquid masks (with column integration). The occurrence fractions were normalized relative to the hourly MPL data availability, under the assumption that the measured period provided an acceptable representation of the whole hour. It should be noted that the MPL

pulse can occasionally be completely attenuated by optically thick cloud layers (for example, as part of a frontal system). Therefore, the real cloud top, geometrical cloud thickness, and potentially, the liquid occurrence are underestimated by the MPL in these situations.

**3 Polar WRF simulations**

WRF is an extensively used community numerical weather prediction model for numerous applications world-wide (e.g.,

Skamarock et al., 2008). Most of the polar optimizations for Polar WRF are added in the Noah LSM (Barlage et al., 2010) and improve the representation of heat transfer through snow and ice (Hines and Bromwich, 2008; Hines et al., 2015). Fractional sea ice was implemented in Polar WRF by Bromwich et al. (2009), followed by the addition of specified variable sea ice thickness, snow depth on sea ice, and sea ice albedo. These updated options were developed by the Polar Meteorology Group (PMG) at Ohio State University's Byrd Polar and Climate Research Center and were included in the standard release of WRF

(https://www.mmm.ucar.edu/weather-research-and-forecasting-model) with the help of the Mesoscale and Microscale Meteorology Division at NCAR (Hines et al., 2015). Hines et al. (2011) made comparisons for cloud and radiation quantities between Polar WRF 3.0.1.1 simulations and observations at the North Slope of Alaska ARM site.

Recently, Deb et al. (2016) evaluated Polar WRF 3.5.1 versus near-surface observations from West Antarctica. They found

that pressure is simulated with high skill, and wind speed is generally well represented. The timing and amplitude of strong wind events were well captured. There were weaknesses in the diurnal cycle of temperature, especially denoted by a cold summertime minimum temperature bias. This was attributed to a negative bias in downwelling longwave radiation, consistent with clouds over Antarctica being poorly represented by models (e.g., Bromwich et al., 2012, 2013b; King et al., 2015; Listowski and Lachlan-Cope, 2017). Arctic modelling studies, however, suggest reason for optimism as Hines and Bromwich

(2017) improved the representation of low-level liquid clouds by Polar WRF 3.7.1 with adjustments to the microphysics for simulations of the Arctic Summer Cloud-Ocean Study (ASCOS) near the North Pole during August-September 2008.



### 3.1 AMPS

A goal for the AWARE project is to evaluate and improve the numerical weather prediction for Antarctica, where the sparse observational network, the physics of the polar atmosphere, and the steep terrain challenge model capabilities (Bromwich et al., 2012). The critical need for accurate weather forecasting to support logistical and scientific activities has been acute since

the earliest Antarctic explorations. To improve forecasting support for the U.S. Antarctic Program, the National Science Foundation's Office of Polar Program initiated the Antarctic Mesoscale Prediction System (AMPS, Powers et al., 2012) in 2000. AMPS is a real-time numerical weather prediction with Polar WRF through a collaboration between the National Center for Atmospheric Research (NCAR) and the PMG.

For the time of the AWARE WAIS case study, the AMPS grid system consists of a series of nested domains with 60 vertical levels between the surface and the model top at 10 hPa. The lowest level is approximately 8 m above the surface, and 12 layers are in the lowest 1 km. The outermost domain had 30 km horizontal resolution and covered Antarctica and much of the Southern Ocean (Fig 1a). Grid 2 had 10 km resolution and covered the Antarctic continent. Four additional higher resolution domains (3.3 km or 1.1 km) covered the Antarctic Peninsula, the South Pole and the region near McMurdo. For the present

study, only Grid 2 fields are used, and results are bilinearly interpolated to WAIS Divide from the four nearest grid points. Lateral boundary conditions for the outer AMPS domain and initial conditions are provided by the Global Forecast System (GFS, NOAA Environmental Modeling Center, 2003), a global forecast system run by the U.S. National Centers for Environmental Prediction, and were updated every 6 hr. The initial fields are enhanced by the assimilation with 3-D variational data assimilation (Barker et al., 2004). Ingested fields include surface data, upper-air soundings, aircraft observations,

geostationary and polar-orbiting satellite atmospheric motion vectors (AMVs), Constellation Observing System for Meteorology, Ionosphere, and Climate (COSMIC) GPS radio occultations, and Advanced Microwave Sounding Unit (AMSU) radiances. Two AMPS forecasts are begun each day starting from GFS analyses at 0000 UTC and 1200 UTC. For the current study we use AMPS output for hours 12 - 21 at 3 hr intervals. Thus, our AMPS fields have a spin-up of a minimum of 12 hrs, with the possibility of fluctuations every 12 hrs due to the change in initialization time. AMPS forecast fields in original WRF

format are available from http://www.earthsystemgrid.org/project/amps.html. Selected AMPS output fields for March 2006 - December 2016 for Grids 2-6 can be downloaded from http://polarmet.osu.edu/AMPS/.

The scarcity of Antarctic meteorological observing stations and satellite blackout periods that can coincide with peak flight times increase the need for AMPS accuracy. Wille et al. (2017) note that unpredicted fog, low ceilings, and high winds lead to

costly flight mission failures over Antarctica, thus accurately predicting acceptable flight windows is essential to prevent costly delays for science missions and cargo transportation. Unfortunately, AMPS has been shown to underestimate low clouds over the Antarctica (Wille et al., 2017). According to Pon (2015) the cloud fraction product in AMPS is so unreliable that most





forecasters rely more on AMPS relative humidity as a proxy for cloud predictions. Therefore, addressing the cloud prediction in AMPS is a primary concern of this work.

AMPS simulations used for 0000 UTC 1 December 2015 - 1200 UTC 19 January 2016 employ Polar WRF 3.3.1 as described
by Wille et al. (2017). Afterward, the AMPS forecast system was upgraded to Polar WRF 3.7.1 (Table 1). The update has no impact on our analyses for the WAIS Divide where all of the observations concluded prior to the change. Grid 2 at 10 km resolution has 667 by 628 horizontal grid points. The boundary layer is represented with the Mellor-Yamada-Janjić planetary boundary layer scheme with nonsingular implementation of level-2.5 Mellor-Yamada closure for turbulence in the planetary boundary layer and free atmosphere (Janjć., 1994). The surface physics are represented with the 4-layer Noah land surface
model with polar modifications (Bromwich et al., 2009; Hines et al., 2015). Other physics options include the Goddard shortwave radiation scheme (Chou et al., 2001), and the Rapid Radiative Transfer Model for GCMs (RRTMG, Clough et al., 2005) longwave radiation scheme. The WRF single-moment 5-class scheme (WSM5C, Hong et al., 2004) is employed to represent the cloud microphysics.

### 3.2 Polar WRF 3.9.1 simulations

Additional numerical simulations during the time of the AWARE field program are conducted with Polar WRF version 3.9.1 (Table 1). These are single-domain simulations with the same grid and topography as AMPS grid 2 (Fig. 1b). The 60 vertical layers are identical to the AMPS simulations. In addition to AMPS, we lean upon our experience with polar simulations for the selection of physical parameterizations (e.g., Wilson et al., 2011, 2012; Bromwich et al., 2013b; Cassano et al., 2017; Hines and Bromwich, 2017). The Mellor–Yamada–Nakanishi–Niino (MYNN; Nakanishi and Niino, 2006) level-2.5 scheme is used
for the atmospheric boundary layer and the corresponding atmospheric surface layer. We use RRTMG for longwave and shortwave radiation. Cloud liquid water, cloud ice, and snow impact the shortwave and longwave radiation, but rain water is not used in the radiation calculations. Cumulus is parameterized with the Kain-Fritch scheme (Kain, 2004). The polar-optimized Noah land surface model is also used. Similar to AMPS simulations for grid 2, no nudging is applied for the Polar WRF 3.9.1 simulations. The simulations presented here input fractional sea ice concentrations from gridded fields at 12.5 km
resolution processed by l'Institut Francais de Recherché Pour l'Exploitation de La Mer (ftp://ftp.ifremer.fr/ifremer/). The sea ice fraction for 1200 UTC 10 January 2016 is shown in Fig. 2b. Sea ice albedo is set at 0.80, same as the snow albedo.

As shown in Table 1, four different schemes are employed for the cloud microphysics, as we wish to see which scheme most accurately represents the atmospheric hydrology and cloud radiative impact. Listowski and Lachlan-Cope (2017) previously
tested five schemes with Polar WRF 3.5.1 for simulations over the central Antarctic Peninsula, however, we are interested in two newer advanced schemes that have become available in more recent versions of WRF. Furthermore, WAIS Divide is more southerly, colder, and the local atmosphere is likely to be more pristine than over the Antarctic Peninsula, where the oceanic influence is strong.



First, we consider WSM5C as it is the microphysics scheme used for AMPS. This widely used scheme is computationally efficient and considers cloud water, cloud ice, rain, and snow as hydrometer classes. Cloud water and cloud ice are suspended, while rain and snow gradually precipitate out with a fall speed. Supercooled water is allowed to exist, and falling snow
gradually melts at temperatures above 0°C. Given that the AMPS simulations and the new Polar WRF 3.9.1 simulations are not conducted with identical model configurations, the simulation referred to as WSM5C (Table 1) is required for comparisons.

Three more advanced schemes are also tested. Following Hines and Bromwich (2017), the two-moment Morrison scheme (e.g., Morrison et al., 2005, 2009) is used as it has been extensively tested in the Arctic and known for its ability to simulate
supercooled liquid water (e.g., Morrison et al., 2008; Klein et al., 2009; Solomon et al., 2011; 2014; 2015). It was amongst the best performing schemes in Listowski and Lachlan-Cope's (2017) simulations. This two-moment bulk microphysics scheme predicts mixing ratios for cloud water, cloud ice, rain, snow and graupel and number concentrations for cloud ice, snow, rain and graupel. Particle size distributions are specified with gamma functions. IN are parameterized according to the Cooper curve, with greater ice crystal concentrations at lower temperatures. The prediction of two-moments (number concentration
and condensate mixing ratio) allows a more robust treatment of the particle size distributions that are important for the microphysical process rates and cloud/precipitation evolution. The liquid water droplet concentration for clouds, however, is specified in the WRF implementation. The standard setting with WRF is 250 $cm^{-3}$. Hines and Bromwich (2017) found best results during the pristine ASCOS study in the eastern Arctic when the value was reduced to 20 $cm^{-3}$ or less. For our AWARE simulations, we have selected 50 $cm^{-3}$. The observations of Lachlan-Cope et al. (2016) and O'Shea et al. (2017) suggest liquid
droplet concentrations are typically above 100 $cm^{-3}$ for clouds over the Antarctic Peninsula.

Simulations are also performed with the aerosol-aware Thompson microphysics (Thompson and Eidhammer, 2014) that is an advancement over the earlier Thompson et al. (2008) bulk microphysics scheme that was one-moment for cloud water and two-moment for cloud ice. This microphysics scheme accounts for cloud nucleating aerosol particles and five water species:
Cloud water, cloud ice, rain, snow and graupel. The scheme includes first order aerosol treatment with interactive IN and cloud condensation nuclei (CCN) concentrations that can vary in a storm or a cloud. Nucleation or complete evaporation of hydrometeors deplete or add to condensation nuclei. Cloud water, cloud ice and rain are treated with two-moment predictions, but snow with only single moment (mixing ratio) predictions. We refer to this scheme as the Thompson scheme. All cloud ice with diameters exceeding 200 microns are converted to snow, which tends to reduce cloud ice mixing ratios and ice particle
diameters in comparison to other schemes (Greg Thompson, personal communication, 2017). Monthly global values for water-friendly and ice-friendly aerosols are from a seven-year simulation of the Goddard Chemistry Aerosol Radiation and Transport (GOCART) model.





The final microphysics scheme is the Morrison-Milbrandt P3 scheme (Morrison and Milbrandt, 2015) hereafter called the P3 scheme. The use of the WRF 3.9.1 in our simulations is motivated by the addition of P3 to the microphysics options. The new scheme avoids the arbitrary categorization of frozen hydrometers into cloud and precipitation, and thus allows for a continuum of particle properties. There are four ice mixing ratio variables: total mass, rime mass, rime volume, and number, allowing for

four degrees of freedom. Liquid hydrometers use a standard two-moment approach with cloud and rain categories. The constant liquid droplet number, 400 $cm^{-3}$, is larger than the standard value for the Morrison scheme.

Both the P3 scheme and the Thompson scheme were unavailable in Polar WRF 3.5.1 when Listowski and Lachlan-Cope (2017) ran simulations for the Antarctic Peninsula. They tested the WSM5C, the WRF double moment scheme, the Morrison scheme,

the older Thompson scheme (Thompson et al., 2008), and the Milbrandt scheme (Milbrandt and Yau, 2005). The older Thompson scheme lacks the aerosol predictive ability of the newer Thompson scheme, and is single moment in cloud water. The latter three schemes simulated clouds in best agreement with observations (Listowski and Lachlan-Cope 2017). All schemes were unsuccessful in representing the supercooled water for some temperature ranges, but the results show that more advanced microphysical parameterizations show improvements in representing Antarctic clouds.

Initial and boundary conditions of meteorological fields for Polar WRF 3.9.1 simulations are interpolated from ERA-Interim reanalysis (ERA-I; Dee et al., 2011) fields available every 6 h on 61 sigma levels and the surface at T255 resolution. This differs from the GFS fields used for AMPS simulations. We have made this change to obtain the best available agreement with observed clouds and radiation. Bracegirdle and Marshall (2012) found that ERA-I best represented the atmospheric circulation

near Antarctica among the reanalyses they evaluated. Bromwich et al. (2013b) found that the boundary layer temperature fields were better represented in WRF simulations driven by ERA-I. Nudging toward analysis fields or observations is not performed on grid 2 during the forecast segment of the AMPS simulations, and no nudging is included for the Polar WRF 3.9.1simulations. Besides the microphysics schemes that are of interest to us, some differences between AMPS and Polar WRF 3.9.1 simulations will occur due to the different base versions of WRF, the source for driving initial and boundary conditions,

and the data assimilation used for AMPS initialization. Strict equality between AMPS and Polar WRF simulations is not required for the goals of this paper, as we are interested in testing the sensitivity to the microphysics parameterization.

The five simulations for this study are shown in Table 1. AMPS 3-hr output was retrieved for 1 December 2015 to 31 January 2016. Four Polar WRF 3.9.1 simulations with different microphysics schemes were then performed. AMPS has the same

microphysics as the WSM5C simulation, however, the length of run segments is longer with the Polar WRF 3.9.1 simulations. A minimum of 12-hour spin-up is taken for each segment initialized at 0000 UTC each day for 3 December 2015 to 19 January 2016. Output each hour for hours 12-35 is combined into fields spanning 1200 UTC 3 December 2015 to 1100 UTC January 2016. Polar WRF output is bilinearly interpolated from the four nearest grid points to the location of WAIS Divide.





## 4 Results

The time period of the December 2015-January 2016 field program at WAIS Divide includes a major melting event over the Ross Ice Shelf and the adjacent Siple Coast of West Antarctica (Nicolas et al. 2017). Temperature over the Ross Ice Shelf and West Antarctica increased after 10 January, and many observing sites there experienced maximum temperatures above freezing

for several days during the melting event. The onset of the melting event is demonstrated by Fig 2a which shows the sea level pressure field, 2 m temperature and 10 m wind speed from the WSM5C simulation at 1200 UTC 10 January. Nicolas et al. (2017) discuss the contribution of a blocking high between 90-120°W to the melting event. Fig. 2a displays anticyclonic shear for the wind barbs at this location. Northerly winds produce widespread advection of warm air over the Ross and Amundsen Seas to the ice shelf and West Antarctica.

### 4.1 Temperature and radiation

Time series of the 2-m temperature at WAIS Divide for 7 - 15 January reveal large warming after 1200 UTC 10 January (Fig. 3a). The observed temperature increases by 13.6°C over 10 hours after the minimum, then increases further to -1.4°C at 1800 UTC 11 January. Warmer locations at lower elevations can be inferred to be above the freezing point and experiencing melting. After a second peak of -1.8°C late on 12 January, the WAIS Divide temperature gradually cools. AMPS has a slight cold bias

prior to the warming, then a cold bias of several degrees during the warm period that follows (Fig. 3a). Interestingly, the WSM5C simulation with Polar WRF 3.9.1 driven by ERA-I eliminates most of the cold bias prior to 10 January and during the warm period. The minimum temperature, however, drops to -22.4°C at 0800 UTC on 10 January in WSM5C.

Table 2 shows statistics of simulations compared to observations. 1099 hourly observations are available for most

meteorological variables from 0600 UTC on 4 December to 0000 UTC on 19 January. Only values every 3 hr are used for AMPS statistics, since output was available at these intervals, so means, biases and other statistics are impacted by the reduced number of values (367). For each variable, Table 2 shows observed averages, and the following rows show AMPS, WSM5C, Morrison, Thompson, and P3 statistics. AMPS has a cold bias of 1.6°C during the observed period, and this is reflected in the time series shown in Fig. 3a. A cold bias is still present in WSM5C. However, it is reduced to 0.3°C (Table 2). Both biases are

statistically significant from zero at the 99% confidence level according to the Student's t-test.

The reduced cold bias for WSM5C can be understood following the sensitivity tests by Bromwich et al. (2013b) with driving by the GFS final analysis (FNL) and ERA-I. They found the sensitivity to the source for initial and boundary conditions varied depending upon season and the choice of physical parameterizations. Their comparison using Polar WRF 3.2.1 with the MYNN

PBL and the RRTMG radiation scheme has the closest model configuration to that used for AMPS and the Polar WRF 3.9.1 simulations. They found that the 2 m temperature bias changed from -3.3°C to 0.1°C with the switch from driving by FNL to driving by ERA-I (see their Table 5). Furthermore, the 2 m dewpoint bias increased from 1.2°C to 4.0°C.



The warmer and moister atmosphere in the Polar WRF 3.9.1 simulations is demonstrated by vertical profiles of temperature and specific humidity biases compared to radiosonde observations (Fig. 4). There is a general cold bias, except with the more advanced microphysics schemes near 1900 m above sea level where the biases reach 0.8 to 0.9°C (Fig. 4a). Thus, there is a

weaker near-surface lapse in the simulations than the observations (not shown). The most extreme bias is the near-surface cold bias for AMPS that reaches 2.3°C. The cold bias for AMPS is also larger than 1°C between 3500 and 5100 m ASL.

An especially striking difference between the AMPS simulation forced with GFS and the simulations driven with ERA-I is shown in Fig. 4b. AMPS is dryer than the radiosonde observations at WAIS Divide at all levels shown, especially in the lowest

3000 m ASL. The WSM5C simulation is slightly drier than the other Polar WRF simulations. The simulations with the advanced microphysics are moister than the observations just above the surface with biases as large as 0.13 g kg$^{-1}$. Above the boundary layer, the specific humidity biases are small, generally below as 0.03 g kg$^{-1}$, for the simulations with the advanced microphysics. From Fig. 4, we can attribute the differences between the AMPS and WSM5C simulations to the colder and drier atmosphere initiated with GFS initial conditions for AMPS.

Figures 3b and 5b help to explain the near-surface temperature results. Downwelling longwave radiation shows a clear negative bias for both AMPS and WSM5C, but the magnitude is much larger for the former. Table 3, with contribution from SEBS observations for 7 December to 16 January, shows that the downwelling longwave bias is quite large, -41.5 W m$^{-2}$ for AMPS. The bias is reduced to -14.8 W m$^{-2}$ for WSM5C. The deficit in longwave radiation is driving the cold bias. Even though the

downwelling shortwave biases are positive for AMPS and WSM5C (Table 3), most of the solar flux is reflected as the surface. Thus, the net radiation flux bias is negative, -3.3 W m$^{-2}$ for AMPS. Since a negative bias in downwelling longwave radiation and a positive bias for downwelling shortwave radiation are found for both AMPS and WSM5C, we believe Polar WRF 3.9.1 simulations can be used to explore the biases in AMPS, and to seek improvements. Downwelling and upwelling longwave biases for both AMPS and WSM5C are all statistically significant (Table 3).

Figure 5 shows diurnal cycles of average fields for 2-m temperature, downwelling longwave radiation, downwelling shortwave radiation, and upwelling shortwave radiation. The time periods for averaging are 4 December 2015 – 19 January 2016 for the temperature and 7 December 2015 – 16 January 2016 for the radiation terms. Simulated biases in these fields vary with time of day, with local noon near 1930 UTC. To provide an idea of the statistical significance of differences in Fig. 5a, we use the

Student's t-test for AMPS and the observations. The observed temperature time series was adjusted each hour by a constant value until the statistical significance of the model minus observed difference was at the boundary of the 95% confidence level. Accounting for autocorrelation in the temperature time series, the degrees of freedom was reduced by a factor of 3. Accordingly, the bias at which the statistical confidence would be 95% could be established. The error bars every 3 hrs in Fig. 5a show the range next to the observations for which differences are not statistically significant. Since AMPS values and


observations of the surface energy balance are simultaneously available only 4 times a day, we use the WSM5C simulation and the observations to determine the statistical significance error bars for Figs 5b and 5c (every two hours beginning at 0100 UTC).

The AMPS mean temperature is less than the observed value at all AMPS output times. Only 0300 UTC is not statistically significant. The observations have an earlier minimum of -16.0°C at 0700 UTC, while the AMPS minimum of -18.5°C occurs at 1200 UTC. The AMPS cold bias, peaks at 1200 UTC (3.1°C). For the Polar WRF 3.9.1 runs, WSM5C is close enough to the observations to be within statistical uncertainty for most hours, except near the time of minimum temperature, when there is a cold bias of 1-2°C. The simulations with more advanced microphysics schemes are warmer than the observations during
the hours of decreasing temperature. P3 is warmest during these times with statistically significant biases of 1.1 to 1.7°C. The transition between run segments at 1200 UTC results in a temperature decrease of up to 2°C, but the change is much less for WSM5C. Starting at 1500 UTC, the Polar WRF 3.9.1 simulations show small temperature biases that are not statistically significant. At or just after the time of maximum temperature, the Polar WRF 3.9.1 simulations show increased warm biases that are statistically significant for Morrison, Thompson and P3. Obviously, the choice of microphysics scheme strongly
impacts the temperature bias at WAIS Divide, and this is shown in Table 2 with warm biases of 0.1, 0.5, and 0.7°C using the Morrison, Thompson, and P3 schemes, respectively.

For downwelling shortwave radiation (Fig. 5c), AMPS has statistically significant positive biases at all hours, with the bias peaking at 106 W m$^{-2}$ at 1500 UTC. The bias is much reduced for the Polar WRF 3.9.1 simulations and not statistically
significant at most observation times. The Morrison scheme, however, does show a statistically significant positive bias ahead of solar noon, while P3 shows a negative bias after solar noon. Table 3 shows that the overall biases during the observing period are 70.4, 17.0, 19.8, 2.5, and -14.2 W m$^{-2}$ for AMPS, WSM5C, Morrison, Thompson, and P3, respectively. All these biases are statistically significant at the 99% confidence level, except for the Thompson scheme for which the bias fails the 95% confidence test.

The shortwave results are encouraging and suggest that advanced microphysics schemes can greatly alleviate, and perhaps even reverse Antarctic radiation biases in numerical simulations. It may appear odd, however, that the upwelling shortwave radiation shows negative biases for all the Polar WRF 3.9.1 simulations that do not coincide with downwelling biases. The difference can be explained by the specified snow albedo in the WRF Noah routine. The specified maximum snow albedo is
0.8 for Noah, and average simulation albedos are slightly below this value. The average observed albedo, however, is 0.843. Therefore, a higher fraction of solar insolation is reflected at WAIS Divide than in these simulations. This results in a deficit of upwelling shortwave radiation (Table 3, Fig. 5d). The deficit increases the net radiation and contributes to the warm bias for the Morrison, Thompson and P3. The impact of the albedo can be seen in the slope of the temperature curves after 1200 UTC in Fig. 5a.



.

We ran a sensitivity test with segments initialized at 0000 UTC each day between 6 January and 16 January 2016. The active period for analysis is 1200 on UTC 6 January until 1100 UTC on 17 January. The settings were equal to the WSM5C, however, the albedo over glacial ice was increased to 0.84, closer to the observed albedo at WAIS Divide. For the used part of the

segments (hours 12-35), the 2-m Temperature average was -12.4°C in the sensitivity test. That is, 1.6°C colder than WSM5C during the same period. That is almost twice the magnitude of the spread of the bias in Polar WRF 3.9.1 simulations shown in Table 2. We surmise that a more realistic surface albedo would likely result in a cold bias for the Polar WRF 3.9.1 simulations.

The observed downwelling longwave radiation (see Fig. 5b) has a mean value of 210.6 W m$^{-2}$ (Table 3). AMPS shows a strong

negative bias at all hours that peaks at -53.0 W m$^{-2}$ at 1500 UTC. The magnitude of the bias is much reduced for WSM5C, but the deficit from the observations is statistically significant at the 95% confidence level except at 0300 UTC and 0500 UTC. The overall bias is -14.8 W m$^{-2}$ and is statistically significant at 99% confidence (Table 3). While there is a large difference between AMPS and WSM5C, the microphysics scheme is nevertheless associated with excess incoming shortwave radiation and a deficit in incoming longwave radiation. This is consistent with Listowski and Lachlan-Cope's (2017) WSM5C results

over the Antarctic Peninsula. They also found that more advanced microphysics schemes (the Morrison scheme is the only advanced scheme used by both studies) can alleviate radiation errors. Similarly, the radiation results were improved here with the Morrison, Thompson and P3 schemes. Table 3 shows overall downwelling longwave biases of -7.9, 0.4, and 1.8 W m$^{-2}$ for these schemes, respectively. The latter two biases are not statistically significant from zero. Correspondingly, Fig. 5b shows that the three advanced schemes do not have statistically significant biases at most hours. The Morrison scheme, however,

does show deficits exceeding 14 W m$^{-2}$ at 1300 and 1500 UTC. These longwave and shortwave results suggest strengths and weaknesses in the simulation of Antarctic clouds.

### 4.2 Clouds

Figure 6 shows the average diurnal cycle over 7 December – 17 January of longwave and shortwave cloud forcing at the surface for the simulations. Cloud forcing ($CF$) is defined following Eqn. (1):

$$CF = F_{\text{all sky}} - F_{\text{clear sky}}, \qquad\qquad\qquad\qquad (1)$$

where $F_{all\ sky}$ is the net all sky flux and $F_{clear\ sky}$ is the net clear sky flux that is estimated to occur without the presence of clouds. Cloud forcing represents the warming effect of clouds (or cooling in the case of negative values) and can be calculated for the longwave, shortwave, or combined flux. Pavolonis and Key (2003) used 1985-1993 data including Advanced Very-High-Resolution Radiometer on NOAA polar orbiting satellites and the International Satellite Cloud Climatology Project to

estimate cloud forcing. They found summertime shortwave cloud forcing of about -10 to -18 W m$^{-2}$ for the latitude of WAIS



Divide. Longwave cloud forcing was 17-35 W m$^{-2}$. Polar WRF 3.9.1 produced clear sky flux values for longwave and shortwave radiation, so cloud forcing could be readily calculated. Clear sky shortwave fluxes were not available from AMPS.

Figure 6a clearly shows that the longwave cloud forcing for AMPS is weak, while the longwave cloud forcing for WSM5C is
less than that of the more advanced schemes. The results for AMPS and WSM5C are consistent with the cold biases during these simulations. P3 produces the greatest overall longwave cloud forcing, but the impact varies somewhat with time of day. Thompson produces nearly as much longwave cloud forcing as P3. The overall averages are 12.2, 31.9, 37.1, 44.8, and 46.1 W m$^{-2}$ for AMPS, WSM5C, Morrison, Thompson and P3, respectively. The simulated values can be compared to monthly surface cloud forcing estimated by Scott et al. (2017) from the Clouds and the Earth's Radiant Energy System (CERES)
CALIPSO-CloudSat-CERES-MODIS dataset (Kato et al., 2011). Using 2007-2010 satellite observations for points near WAIS Divide, they found January values of 57.3, -29.1, and 28.3 W m$^{-2}$ for longwave, shortwave, and net cloud forcing, respectively. The simulated cloud forcing tends to be much greater than the climatological values of Pavolonis and Key (2003), yet smaller than Scott et al.'s (2017) values. Given that clouds contributed to the major melting event during January 2016 (Nicolas et al., 2017), cloud forcing in excess of the climatological mean is possible for this month.

Fig. 6b shows shortwave cloud forcing which has a cooling effect on the surface. There are considerable differences between the advanced microphysics schemes. The overall averages are -11.0, -10.1, -13.7, and -18.5 W m$^{-2}$ for WSM5C, Morrison, Thompson and P3, respectively. P3 shows a strong diurnal cycle with a minimum magnitude (-13.5 W m$^{-2}$) at 0800 UTC and a maximum magnitude (-25.2 W m$^{-2}$) at 2300 UTC near the time of maximum insolation and temperature. In contrast, the
Morrison scheme has a minimum magnitude when insolation is large. The more advanced microphysics schemes produce stronger cloud radiative properties than the WSM5C scheme. Of the three advanced schemes, P3 shows the strongest cloud radiative impact, while Morrison shows the least.

The average diurnal cycles of sensible heat flux and the conductive heat flux into the ice at WAIS Divide are shown in Fig. 7.
The conductive flux was not directly measured by Nicolas et al. (2017), however, the flux was estimated from the residual of other terms in the surface energy balance. The diurnal cycle of sensible heat flux was greatly amplified in the simulations compared to the observations (Fig 7a). The positive sensible heat fluxes into the atmosphere are especially large near the time of maximum temperature, with a maximum of 32.3 W m$^{-2}$ at 1900 UTC for P3. The maximum is much smaller for AMPS (15.3 W m$^{-2}$), which is colder. The overall average observed value is small, 0.9 W m$^{-2}$ (Table 3). Modelled overall averages
vary from 1.8 W m$^{-2}$ for AMPS to 11.4 W m$^{-2}$ for P3.

The conductive flux into the ice is a critical term for mass balance of West Antarctica. Therefore, it is important for modelling studies to be able to well represent this quantity. Positive values are expected during December and January when insolation is large. The overall average for the residual estimate of Nicolas et al. (2017) is 7.5 W m$^{-2}$ during the observational period



(Table 3). AMPS, which has a cold bias, also has a bias of -3.2 W m$^{-2}$ for the conductive flux. The overall biases are positive for all the Polar WRF 3.9.1 simulations, with values of 2.2, 2.1, 2.8, and 5.1 W m$^{-2}$ for WSM5C, Morrison, Thompson, and P3, respectively. The large values during the warmer part of the day are key to the positive biases (Fig. 7b).

While the previous analysis has concentrated on radiation fields and the surface energy balance, we now more directly examine the observed and simulated clouds. Figure 8 shows the LWP for 2 - 18 January 2016. The uncertainty of observed LWP is 10 g m$^{-2}$ (Cadeddu et al., 2009). Modelled LWP includes both suspended liquid cloud droplets and falling rain. LWP values above 0 are observed at most times, but AMPS and WSM5C simulate non-zero values only during 11-12 January (Fig. 8a). The results demonstrate the known difficulty of the WSM5C microphysics to simulating liquid water for polar clouds (e.g.,
Listowski and Lachlan-Cope, 2017). The more advanced microphysics schemes simulate liquid water much more frequently than WSM5C, but do not well represent the instantaneous observed liquid water (Fig. 8b). Therefore, we suggest that the simulation of liquid water in polar clouds remains problematic (e.g., King et al., 2015; Hines and Bromwich, 2017; Listowski and Lachlan-Cope, 2017).

Table 4 shows the average condensate over 0000 UTC 2 January to 0000 UTC 18 January. The average observed LWP, 23 g m$^{-2}$, is larger than in any of the simulations. The largest simulated value is 15.5 g m$^{-2}$ for P3, consistent with magnitude of cloud forcing for this simulation (Fig. 6). Morrison has smaller LWP, 5.1 g m$^{-2}$, than Thompson or P3, corresponding to the weaker cloud forcing in Fig. 6. LWP is small, 0.43 g m$^{-2}$ and 0.88 g m$^{-2}$, respectively for AMPS and WSM5C. The radiative impact of microphysics schemes for WAIS appears to be strongly linked to the ability to simulate liquid water.

Caution should be applied in comparing the distributions of suspended and precipitation hydrometers between schemes since the definitions of such categories are arbitrary and poorly defined physically (Morrison and Milbrandt, 2015). The distribution of hydrometers can be helpful, however, in understanding the inner workings of a microphysics scheme and comparing the simulated amounts of liquid and ice. Simulated cloud water tends to be an order of magnitude or two larger than rain water.
Little ice is simulated as graupel or rime. Morrison simulates an order of magnitude more snow than cloud ice, while the difference is two orders of magnitude for Thompson. In contrast, the simulations with the WSM5C microphysics produced high amounts of cloud ice but little amounts of snow. The total ice condensate in the WSM5C simulation, 21 g m$^{-2}$, is more than twice the value for AMPS, 10 g m$^{-2}$. More cloud ice in WSM5C can explain the greater cloud radiative impact compared to AMPS given that liquid water is rarely present (Figs. 6a and 8a). For the more advanced microphysics schemes, ice water
path (IWP) varies from 15 g m$^{-2}$ for Morrison to 23 g m$^{-2}$ for Thompson and P3. Fig 7c indicates that the time series of IWP often show a rough similarity between schemes. Accordingly, the amount of liquid water appears to be a stronger factor in the difference between simulations results.





Figure 9 shows times series of cloud occurrence fraction at the WAIS Divide during the MWR availability period. Figure 9a

shows values determined from the MPL observations. For the simulations, we use the model cloud fraction formulation of

Fogt and Bromwich (2008) calibrated to manual McMurdo cloud fraction observations:

$$Cloud\ Fraction = 0.075\ LWP + 0.170\ IWP\ , \qquad\qquad (2)$$

where the total cloud fraction, $Cloud\ Fraction$ is based upon the LWP and IWP in g m$^{-2}$. The cloud fraction is limited to the

maximum value of 1. Cloud occurrence fraction from the MPL is not identical to standard observer-based cloud fraction

observations (e.g., Wagner and Kleiss, 2016). However, the model cloud fraction by Eq. (2) is typically one or very close to

zero, so the effective differences between cloud fraction and cloud fraction occurrence is minimized for comparisons between

model and observations.

The observed cloud occurrence fraction is frequently 1, and the average is 0.77 during this time (Fig. 8a). Cloud free times are

more common for AMPS, thus the average is 0.32 (Fig. 8b). The Polar WRF 3.9.1 simulations show some similarity in their

time series of cloud fraction, with average varying from 0.59 for WSM6C to 0.71 for P3. Microphysics schemes with stronger

cloud radiative forcing have larger average total cloud fraction (Figs. 6 and 9).

Liquid cloud occurrence fraction is shown in Fig. 10. Only the first term on the right-hand side of Eq. (2) is used to define

modelled liquid cloud fraction. Liquid clouds are frequently observed but are rarely simulated by AMPS (Fig. 10a). The

Morrison scheme simulates liquid clouds much more frequently than WSM5C, but not as frequently as the observations. The

Thompson and P3 schemes simulate liquid clouds more frequently than the Morrison scheme. Average liquid cloud fractions

are 0.65, 0.01, 0.05, 0.20, 0.26 and 0.34 for the observations, AMPS. WSM5C, Morrison, Thompson, and P3, respectively.

Figure 11 shows the vertical distribution of cloud fraction. While the observed cloud fraction is again determined by surface-

based MPL observations, Eq. (2) is inappropriate for point values of cloud fraction in a column. We select the mixing ratio

0.001 g kg$^{-1}$ as the classic WRF minimum hydrometer threshold for cloud in the simulations. Model fraction is either 0 or 1

for total condensate concentrations below or above the threshold.

Remote sensing at WAIS Divide detects clouds that are frequently present below 650 hPa (Fig. 11a). Detectable clouds

decrease with height and are rarely observed above 500 hPa, but this is likely due to attenuation of the lidar pulse at lower

altitudes. The simulated clouds are much deeper and frequently extend above 400 hPa (Fig. 11b-f). The minimum threshold

of 0.001 g kg$^{-1}$ allows clouds with the density of very thin cirrus that may be difficult to observe. We found that simulated

cloud tops (not shown) are sensitive to the specification of the threshold.



Figure 12 shows liquid cloud occurrence fraction to be more confined to the lower troposphere than total cloud occurrence fraction (Fig. 11). The simulated liquid clouds, when present, are near the surface for the simulations with WSM5C microphysics (Figs. 12b and 12c). The more advanced microphysics schemes simulate deeper liquid clouds than are observed by MPL.

Figure 13 shows the mean cloud fraction profiles above ground level (AGL) for 2 January to 16 January. As noted earlier, the MPL pulse attenuation likely results in some underestimation of both the total cloud and liquid occurrence fractions at higher elevations. Returning to Fig. 11 that shows shallow clouds with variable vertical structure observed by the MPL, while the simulations have deep, vertically aligned clouds, the means shown in Fig. 13 reflect this difference in vertical structure. The

averaging of frequent deep cloud structures results in high mean values for the simulations, compared to the means of the more variable observations. Therefore, a vertically aligned cloud overlap better represents the simulated clouds than a random overlap. These stacked clouds reduce the modelled cloud fraction shown in Fig 9, as the middle cloud layer is on of top of the low cloud layer, rather than additive to the cloud fraction. The observed average total cloud fraction peaks at 0.51 at 185 m AGL (Fig 13a). The fraction decreases to 0.30 near 500 m AGL then 0.10 above 1500 m. The profiles suggest that there could

be slightly elevated (liquid-bearing) cloud occurrence at 2135 m AGL. The observed liquid cloud fraction is more surface-based with a peak of 0.28 at both 115 and 185 m AGL, and decreasing values to 0.06 at 410 m (Fig 13a).

The simulated cloud fraction profile peaks near the surface for AMPS and WSM5C (Fig. 13b). For AMPS (WSM5C), the maximum is 0.50 (0.64) at 8 m AGL (84 m). Cloud fraction is higher for the advanced microphysics schemes, with all having

maxima above 0.64 at heights below 600 m AGL. Largest cloud fraction is 0.69 at 365 m for Thompson. After decreasing with height up to 2000 m, simulated cloud fraction tends to stabilize with height above that. Differences between the simulated and observed cloud fraction profiles in Figs. 13b may be linked with the near-surface temperature bias shown in Fig. 4a.

The mean simulated liquid and ice cloud fractions are shown in Fig. 14. The values are from 2-16 January 2016, the same

period used for the profiles in Fig. 4. The fractions are based upon the total liquid or ice content. P3 has a unique liquid profile that peaks at 0.26 at 576 m AGL. The Morrison and Thompson simulations have similar liquid cloud fraction profiles with double peaks between 360 and 700 m. The average profiles suggest frequent liquid cloud tops near 1000 m AGL. Liquid cloud fraction for WSM5C peaks at 0.05 at 464 m AGL. AMPS has a peak of just 0.017 in the lowest 50 m. The simulations with advanced microphysics schemes show a reflection of the observed secondary liquid peak above 2100 m AGL (Fig. 14a). Fig.

14b shows that ice is frequently present in the lowest 1000 m of the simulations. All the simulations show maxima for ice in the lowest 500 m varying from 0.50 at 8 m for AMPS and 0.49 for P3 at 365 m to 0.64 for 85 m with WSM5C. The Polar WRF 3.9.1 simulations produce more ice cloud fraction than AMPS. It may be inferred from Fig. 14b that the higher clouds in the simulations are likely to be thin cirrus (ice) clouds.



A sensitivity test referred to as P3-50, was based upon P3 to see if the setting of 400 cm$^{-3}$ for the liquid droplet number concentration had an important impact on results of that simulation. We set the liquid concentration at 50 cm$^{-3}$ in the sensitivity test, same as in the simulation with the Morrison microphysics. We use 1200 UTC 6 January – 1100 UTC 17 January 2016 as the active period for test results. P3-50 exhibited a reduction of the average LWP from 21 g m$^{-2}$ to 16 g m$^{-2}$, compared to the

parent simulation P3. The ice water path is less impacted and reduced by less than 7%. Figure 15 shows the 2-m temperature and surface downwelling shortwave and longwave radiation. The change in specified liquid concentration has small impact on the 2 m temperature, with the largest impact after 10 January when more noticeable amounts of liquid water were simulated (Figs. 8b and 15a). The average temperature in P3-50 (-9.6°C) is the same as in P3 over the test period. The downwelling shortwave radiation, however, is modified with the local noon on January 11, 14 and 15 showing insolation increases of 50-

170 W m$^{-2}$ (Fig. 15b). P3-50 is an improvement on these days. The impact on the downwelling longwave radiation is much smaller (Fig. 15c). Overall, P3-50 has a net increase (decrease) in 23.9 (2.6) W m$^{-2}$ in downwelling shortwave (longwave) radiation compared to P3. Since most of the shortwave radiation is reflected off the Antarctic surface, the net impact on the near-surface temperature is small (Fig. 15a).

## 5 Summary and conclusions

The recent 2015-2017 AWARE field program provides a highly detailed set of remote sensing and surface observations that can be used to study the simulation Antarctic clouds and the surface energy budget. We focus on the December 2015 - January 2016 test period when observations were taken at WAIS Divide. These observations are used for comparison with the AMPS forecasting system and new simulations with Polar WRF 3.9.1. AMPS uses the WRF Single-Moment 5-Class microphysics, while the new Polar WRF 3.9.1 simulations are run with WSM5C and three more advanced microphysics schemes. These are

the Morrison 2-moment microphysics, the Thompson-Eidhammer aerosol-aware microphysics, and the new Morrison-Milbrandt P3 microphysics.

AMPS simulates few liquid hydrometers during austral summer at WAIS Divide, even though liquid clouds are frequently observed by the MPL, primarily a consequence of the WSM5C microphysics in AMPS. Consequently, downwelling shortwave

radiation is excessive at the surface, while downwelling longwave radiation is too small. The WSM5C simulation with Polar WRF 3.9.1 has reduced biases of the same sign. The decreased magnitude in WSM5C appears due to GFS-forcing of initial and boundary conditions for AMPS and while ERA-I is used for WSM5C. Simulated hydrometers are overwhelmingly composed of ice with WSM5C.

The more advanced microphysics schemes show considerable improvement in the simulation of overall cloud fraction, liquid hydrometers, and cloud radiative effects. The instantaneous simulation of liquid remains somewhat problematic even given the improvements. The Morrison scheme simulates less LWC and weaker cloud radiative forcing than the Thompson and P3 schemes. P3 simulates the greatest LWC and cloud radiative effect. All schemes appear to underestimate total cloud fraction





and liquid cloud fraction at the WAIS Divide. The vertical distribution of simulated cloud properties differs from observed profiles, with deeper clouds simulated than observed, although the MPL may not detect the upper regions of clouds due to attenuation.

The work presented here may contribute to improvements to the AMPS simulations of clouds being sought by NCAR if computational efficiency can be achieved (Jordan Powers, personal communication 2018). Moreover, the more extensive AWARE cloud observations at McMurdo over the full seasonal cycle will provide a basis for sensitivity tests designed to seek Antarctic optimizations to the advanced microphysics schemes used for the WAIS Divide. In particular, we plan to work with two more advanced implementations of the P3 microphysics (Milbrandt and Morrison, 2016). Sensitivity tests will also vary

the background IN concentrations in simulations with the Thompson microphysics, as the limited observational evidence suggests that the contributing aerosol concentrations may vary or are unknown over a range of orders of magnitude.

**6 Code availability**

The standard release of WRF can be downloaded from NCAR (https://www.mmm.ucar.edu/weather-research-and-forecasting-model). The polar optimizations can be requested from http://polarmet.osu.edu/PWRF/registration.php.

**7 Data availability**

All the observations from the AWARE field campaign (including the reprocessed MPL data set) can be downloaded from the ARM Data Discovery website (http://www.archive.arm.gov/discovery/). AMPS forecast fields in original WRF format are available from http://www.earthsystemgrid.org/project/amps.html. Selected AMPS output fields for March 2006 - December 2016 for Grids 2-6 can be downloaded from http://polarmet.osu.edu/AMPS/.

**8 Team list**

Keith M. Hines

David H. Bromwich

Sheng-Hung Wang

Israel Silber

Johannes Verlinde

Dan Lubin

**9 Author contribution**

Keith Hines was the primary author and coordinated with the other authors. He conducted the Polar WRF simulations, downloaded and processed AWARE data, and processed the model data.



David Bromwich coordinated the Ohio State component of AWARE and was the primary co-author. He read and provided input on all drafts of the manuscript and helped plan the simulations.

Sheng-Hung Wang processed the AMPS data for the manuscript.

Isreal Silber processed the MPL and cloud mask data. He provided advice on the use of these data.

5 Johannes Verlinde coordinated the Penn State component of AWARE. He provided advice on Antarctic clouds and the AWARE data.

Dan Lubin was the overall coordinator of AWARE. He provided advice on Antarctic clouds and the AWARE project and helped to coordinate the use of CERES data.

## 10 Competing interests

10 The authors declare that they have no conflict of interest.

## 11 Disclaimer

Any opinions presented here are those of the manuscript authors alone and are not necessarily those of Atmospheric Chemistry and Physics.

## 12 Acknowledgments

15 This research is supported by DOE Grant DE-SC0017981 and NSF Grant PLR 1443443. Numerical simulations were performed on the Intel Xeon cluster at the Ohio Supercomputer Center, which is supported by the State of Ohio. We thank Julien Nicolas for providing rawinsonde, surface energy balance and LWP observations for WAIS Divide and Ryan Scott for providing cloud forcing derived from CERES. All the observations from the AWARE field campaign (including the reprocessed MPL data set) can be downloaded from the ARM Data Discovery website 20 (http://www.archive.arm.gov/discovery/). This is Contribution XXXX of the Byrd Polar & Climate Research Center.

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



| Table 1 *Simulations for 3 December 2015 to 21 January 2016* | | | | | |
|---|---|---|---|---|---|
| Run | Model | Microphysics Scheme | Frequency | Start time | Segment Length |
| AMPS | Polar WRF* 3.31/3.7.1** | WRF Single-Moment 5 Class (WSM5C) | 12 hrs | 0000 and 1200 UTC | 24 hr |
| WSM5C | Polar WRF 3.9.1 | WRF Single-Moment 5 Class (WSM5C) | 24 hrs | 0000 UTC | 48 hr |
| Morrison | Polar WRF 3.9.1 | Morrison 2-Moment (Morrison) | 24 hrs | 0000 UTC | 48 hr |
| Thompson | Polar WRF 3.9.1 | Thompson-Eidhammer (Thompson) | 24 hrs | 0000 UTC | 48 hr |
| P3 | Polar WRF 3.9.1 | Morrison-Milbrandt (P3) | 24 hrs | 0000 UTC | 48 hr |

*Antarctic adaptations and data assimilation are included in the AMPS simulation.

**AMPS was upgraded from Polar WRF v. 3.3.1 to v. 3.7.1 on 19 January 2016.



| Table 2 *Model performance at WAIS for 4 December 2015 – 19 January 2016.*\* | | | | | | | |
|---|---|---|---|---|---|---|---|
| Variable | Run | Values | Average | Bias** | Correlation | Mean Absolute Error | Root Mean Square Error |
| Surface Pressure (hPa) | Observation | 1099 | 786.4 | - | - | - | - |
| | AMPS | 367 | 787.2 | 0.9 | 0.99 | 1.2 | 1.5 |
| | WSM5C | 1099 | 787.1 | 0.7 | 0.99 | 1.0 | 1.3 |
| | Morrison | 1099 | 787.1 | 0.8 | 0.99 | 1.0 | 1.3 |
| | Thompson | 1099 | 787.1 | 0.7 | 0.99 | 1.0 | 1.3 |
| | P3 | 1099 | 787.1 | 0.7 | 0.99 | 1.0 | 1.3 |
| 2-m Temperature (°C) | Observation | 1099 | -14.1 | - | - | - | - |
| | AMPS | 367 | -15.7 | -1.6 | 0.86 | 2.1 | 2.7 |
| | WSM5C | 1099 | -14.0 | -0.3 | 0.89 | 1.6 | 2.2 |
| | Morrison | 1099 | -14.4 | 0.1 | 0.91 | 1.5 | 2.0 |
| | Thompson | 1099 | -13.6 | 0.5 | 0.90 | 1.5 | 2.1 |
| | P3 | 1099 | -13.4 | 0.7 | 0.90 | 1.6 | 2.1 |
| 2-m Specific Humidity (g kg$^{-1}$) | Observation | 1099 | 1.23 | - | - | - | - |
| | AMPS | 367 | 1.25 | **0.02** | 0.83 | 0.25 | 0.36 |
| | WSM5C | 1099 | 1.40 | 0.18 | 0.90 | 0.26 | 0.34 |
| | Morrison | 1099 | 1.45 | 0.22 | 0.93 | 0.26 | 0.34 |
| | Thompson | 1099 | 1.47 | 0.25 | 0.92 | 0.28 | 0.36 |
| | P3 | 1099 | 1.49 | 0.26 | 0.92 | 0.29 | 0.37 |
| 10-m Wind Speed (m s$^{-1}$) | Observation | 1099 | 6.0 | - | - | - | - |
| | AMPS | 367 | 5.6 | -0.4 | 0.77 | 1.5 | 2.0 |



| | | | | | | | |
|---|---|---|---|---|---|---|---|
| | WSM5C | 1099 | 5.8 | -0.3 | 0.80 | 1.3 | 1.7 |
| | Morrison | 1099 | 5.9 | -0.2 | 0.79 | 1.3 | 1.8 |
| | Thompson | 1099 | 6.0 | **0.0** | 0.79 | 1.3 | 1.8 |
| | P3 | 1099 | 6.0 | **-0.1** | 0.77 | 1.4 | 1.9 |
| 10-m | Observation | 1099 | 138.3 | - | - | - | - |
| Direction | AMPS | 367 | 137.6 | **-1.6** | 0.58 | 27.4 | 45.2 |
| (degree) | WSM5C | 1099 | 147.8 | 9.5 | 0.67 | 25.3 | 42.9 |
| | Morrison | 1099 | 150.0 | 11.8 | 0.66 | 25.4 | 42.3 |
| | Thompson | 1099 | 151.9 | 13.7 | 0.68 | 26.3 | 43.5 |
| | P3 | 1099 | 154.5 | 16.2 | 0.70 | 26.6 | 43.8 |

*Statistics are calculated from hourly values during 0600 UTC 4 December 2015 – 0000 UTC 19 January 2016 for the observations and the Polar WRF 3.9.1 runs. Values every 3 hours are used for the AMPS results. Six values are shown in each column for each variable. The values are for Observations (average only), Antarctic Mesoscale Prediction System (AMPS,) Polar WRF 3.9.1 simulation with the WRF Single-Moment 5-Class microphysics (WSM5C), Polar WRF 3.9.1 simulation with the Morrison Microphysics (Morrison), Polar WRF 3.9.1 simulation with the aerosol-aware Thompson microphysics (Thompson), and  Polar WRF 3.9.1 simulation with the Morrison-Milbrandt microphysics (P3).

**All biases are statistically significant from zero at the 95% confidence level according to the student's T-test except for values shown in bold. Most biases are also significant at the 99% confidence level.



| Table 3 *Surface Energy Balance at WAIS for 7 December 2015 – 16 January 2016.** | | | | | | |
|---|---|---|---|---|---|---|
| Variable | Run | Values | Average | Bias** | Correlation | Mean Error | RMSE |
| Downwelling Shortwave Radiation | Observation | 492 | 373.1 | - | - | - | - |
| | AMPS | 164 | 446.9 | 70.4 | 0.92 | 76.3 | 97.3 |
| | WSM5C | 492 | 390.1 | 17.0 | 0.94 | 46.9 | 60.8 |
| | Morrison | 492 | 392.9 | 19.8 | 0.94 | 46.5 | 59.5 |
| | Thompson | 492 | 375.6 | **2.5** | 0.94 | 42.4 | 54.8 |
| | P3 | 492 | 358.9 | -14.2 | 0.92 | 51.9 | 67.4 |
| Downwelling Longwave Radiation | Observation | 492 | 210.6 | - | - | - | - |
| | AMPS | 164 | 169.6 | -41.5 | 0.51 | 43.7 | 54.1 |
| | WSM5C | 492 | 195.8 | -14.8 | 0.64 | 26.5 | 35.1 |
| | Morrison | 492 | 202.7 | -7.9 | 0.72 | 22.6 | 29.3 |
| | Thompson | 492 | 211.0 | **0.4** | 0.72 | 21.0 | 27.8 |
| | P3 | 492 | 212.4 | **1.8** | 0.73 | 22.4 | 29.0 |
| Upwelling Shortwave Radiation | Observation | 492 | 314.7 | - | - | - | - |
| | AMPS | 164 | 357.5 | 40.3 | 0.94 | 50.7 | 63.9 |
| | WSM5C | 492 | 310.1 | -4.6 | 0.95 | 38.4 | 46.6 |
| | Morrison | 492 | 313.3 | **-1.4** | 0.95 | 34.1 | 42.4 |
| | Thompson | 492 | 298.7 | -16.0 | 0.96 | 35.4 | 45.4 |
| | P3 | 492 | 277.6 | -37.1 | 0.93 | 51.2 | 65.7 |
| Upwelling Longwave Radiation | Observation | 492 | 256.5 | - | - | - | - |
| | AMPS | 164 | 248.7 | -8.0 | 0.81 | 10.3 | 13.7 |
| | WSM5C | 492 | 254.3 | -2.2 | 0.85 | 8.4 | 11.2 |
| | Morrison | 492 | 257.2 | **0.8** | 0.90 | 7.3 | 9.5 |



|  |  |  |  |  |  |  |  |
|---|---|---|---|---|---|---|---|
|  | Thompson | 492 | 259.8 | 3.4 | 0.89 | 7.3 | 9.6 |
|  | P3 | 492 | 261.1 | 4.7 | 0.89 | 7.9 | 10.2 |
| Net Radiation | Observation | 492 | 12.6 | - | - | - | - |
|  | AMPS | 164 | 10.2 | -3.3 | 0.70 | 15.1 | 18.4 |
|  | WSM5C | 492 | 21.5 | 8.9 | 0.70 | 16.1 | 20.5 |
|  | Morrison | 492 | 25.1 | 12.6 | 0.73 | 17.5 | 22.4 |
|  | Thompson | 492 | 28.0 | 15.5 | 0.75 | 18.7 | 23.7 |
|  | P3 | 492 | 32.6 | 20.0 | 0.72 | 22.5 | 28.4 |
| Sensible Heat | Observation | 492 | 0.9 | - | - | - | - |
| Flux | AMPS | 164 | 1.8 | **0.7** | 0.76 | 5.8 | 7.7 |
|  | WSM5C | 492 | 6.2 | 5.3 | 0.78 | 8.9 | 12.0 |
|  | Morrison | 492 | 8.3 | 7.4 | 0.81 | 10.1 | 13.4 |
|  | Thompson | 492 | 9.7 | 8.8 | 0.83 | 10.6 | 14.2 |
|  | P3 | 492 | 11.4 | 10.5 | 0.81 | 12.1 | 16.0 |
| Latent Heat | Observation | 492 | 4.2 | - | - | - | - |
| Flux | AMPS | 164 | 3.4 | -0.8 | 0.82 | 2.8 | 3.5 |
|  | WSM5C | 492 | 5.7 | 1.5 | 0.81 | 3.6 | 5.1 |
|  | Morrison | 492 | 7.3 | 3.2 | 0.78 | 4.9 | 7.0 |
|  | Thompson | 492 | 8.1 | 4.0 | 0.81 | 5.0 | 7.1 |
|  | P3 | 492 | 8.8 | 4.6 | 0.80 | 5.6 | 7.6 |
| Heat Flux | Observation | 492 | 7.5 | - | - | - | - |
| into the Ice | AMPS | 164 | 5.0 | -3.2 | 0.38 | 9.2 | 11.5 |
|  | WSM5C | 492 | 9.6 | 2.2 | 0.40 | 7.9 | 10.0 |
|  | Morrison | 492 | 9.5 | 2.1 | 0.43 | 7.5 | 9.7 |
|  | Thompson | 492 | 10.2 | 2.8 | 0.42 | 7.9 | 10.0 |
|  | P3 | 492 | 12.6 | 5.1 | 0.47 | 8.5 | 10.7 |





*Statistics are calculated from values every other hour during 0100 UTC 7 December 2015 –
2300 UTC 16 January 2016 for the observations and the Polar WRF 3.9.1 runs. Values at 0300,
0900, 1500, and 2100 UTC used for the AMPS results. Six values are shown in each column for
each variable. The values are for observations (average only}, AMPS, WSM5C, Morrison,
Thompson, and P3.

**All biases are statistically significant from zero at the 95% confidence level according to the
student's t-test except for values shown in bold. Most biases are also significant at the 99%
confidence level.





| Table 4 *Mean Hydrometers (g m⁻² ) at WAIS for 2-18 January 2016* | | | |
|---|---|---|---|
| Liquid Water Path | Observations | | 23.50 |
| | AMPS | | 0.43 |
| | | Cloud Water | 0.42 |
| | | Rain Water | 0.08 |
| | WSM5C | | 0.88 |
| | | Cloud Water | 0.87 |
| | | Rain Water | 0.01 |
| | Morrison | | 5.14 |
| | | Cloud Water | 5.06 |
| | | Rain Water | 0.08 |
| | Thompson | | 6.97 |
| | | Cloud Water | 6.82 |
| | | Rain Water | 0.15 |
| | P3 | | 15.52 |
| | | Cloud Water | 15.34 |
| | | Rain Water | 0.18 |
| Ice Water Path | AMPS | | 10.27 |
| | | Cloud Ice | 10.05 |
| | | Snow Ice | 0.22 |
| | WSM5C | | 20.73 |
| | | Cloud Ice | 19.71 |
| | | Snow Ice | 1.02 |
| | Morrison | | 15.30 |
| | | Cloud Ice | 1.76 |
| | | Snow Ice | 13.54 |
| | | Graupel | 0.001 |
| | Thompson | | 23.42 |
| | | Cloud Ice | 0.41 |
| | | Snow Ice | 22.90 |
| | | Graupel | 0.12 |
| | P3 | | 23.15 |
| | | Cloud Ice | 22.69 |
| | | Snow Ice | 0.46 |
| * Equivalent liquid water depth for hydrometers in mm at WAIS Divide. | | | |




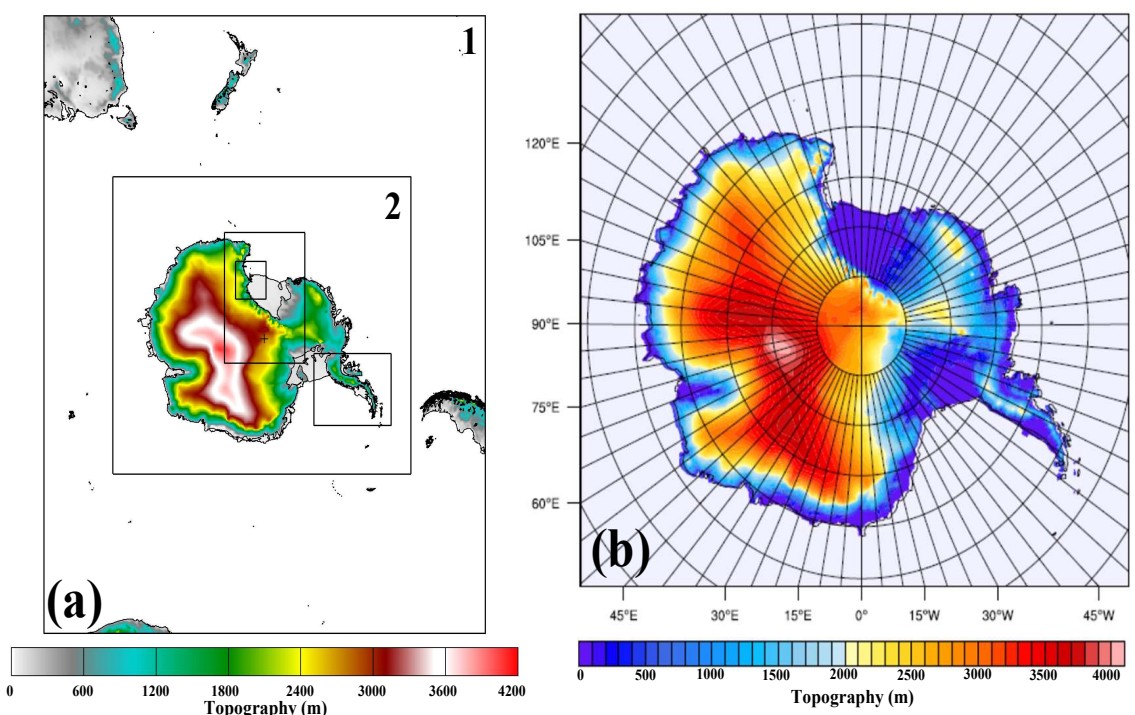

**Figure 1: Antarctic Mesoscale Prediction System (AMPS) grids (a) and grid for Polar WRF 3.9.1 simulations (b). Topography (m) is shown by color scales for both panels. Grid 2 in (a) with 10-km horizontal resolution is the same as the grid shown in (b). Grid 1 in (a) has 30-km resolution.**



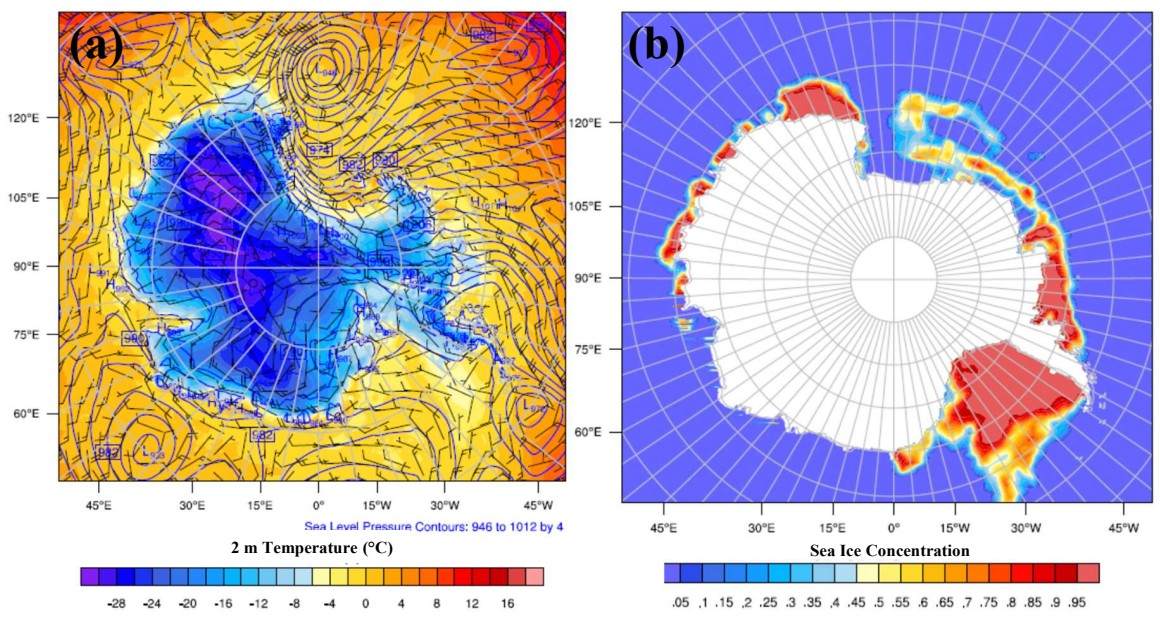

**Figure 2: Plots of (a) simulated 2 m temperature (°C, color scale), sea level pressure (contours, hPa) and 10 wind barbs (m s⁻¹), and (b) sea ice fraction for 1200 UTC 10 January 2016.**





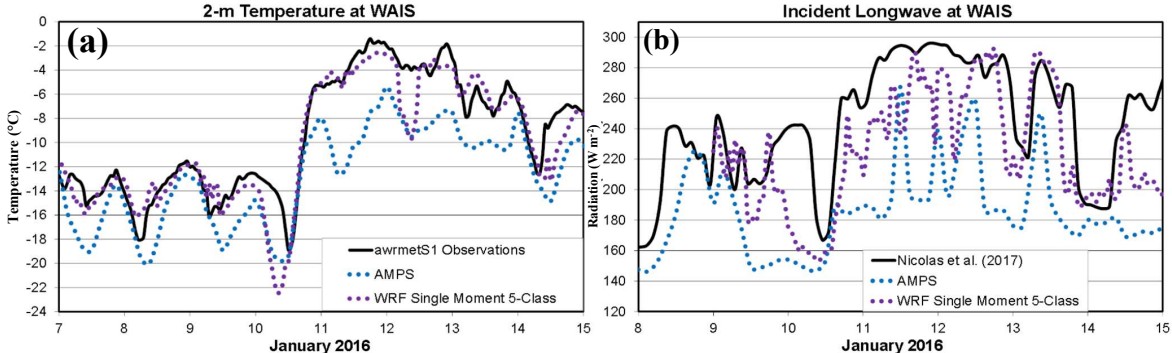

**Figure 3: Time series of (a) 2 m temperature (°C) and (b) downwelling longwave radiation (W m⁻²) for 7-15 January 2016. The solid curves show the observed temperature in (a) and the observed longwave radiation from Nicolas et al. (2017) in (b). AMPS values are shown by dotted blue curves while dotted violet curves show the values from the Polar WRF 3.9.1 simulation with the WRF single-moment 5-class microphysics.**





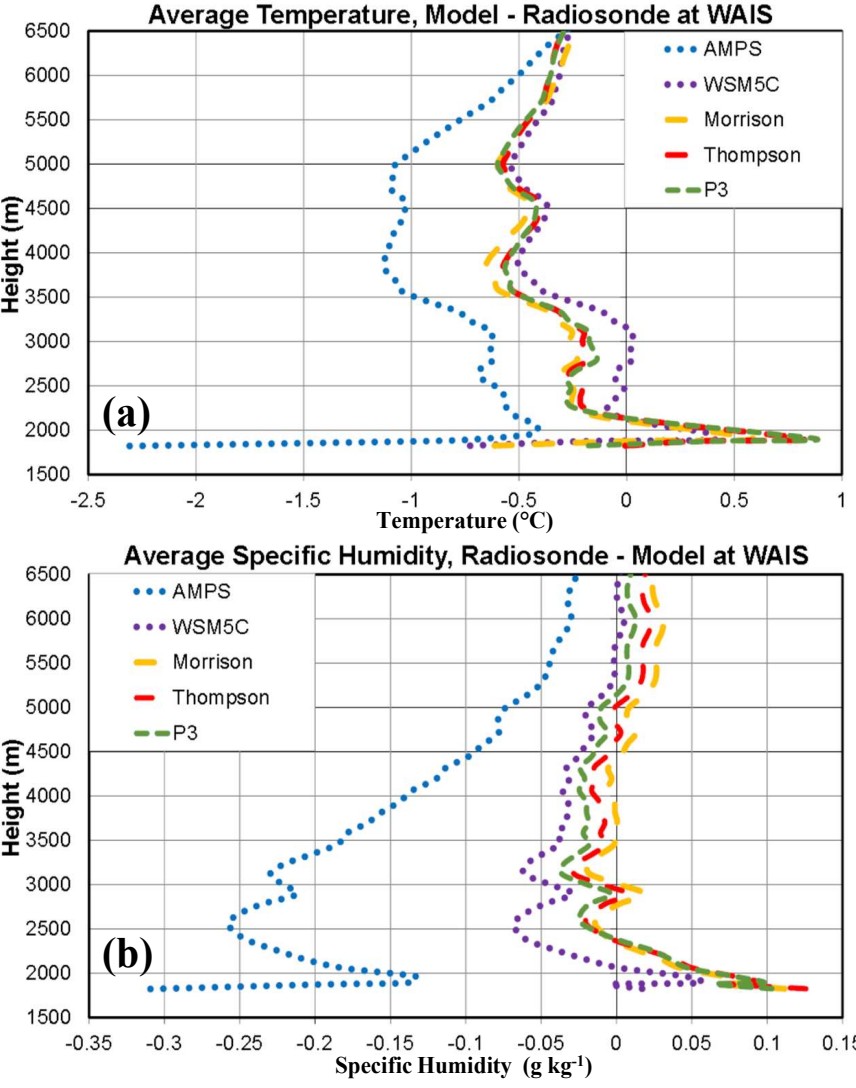

**Figure 4: Vertical profiles of average (a) temperature (°C) and
(b) specific humidity (g kg⁻¹) differences between simulations
and radiosonde observations over 2–16 January 2016.**




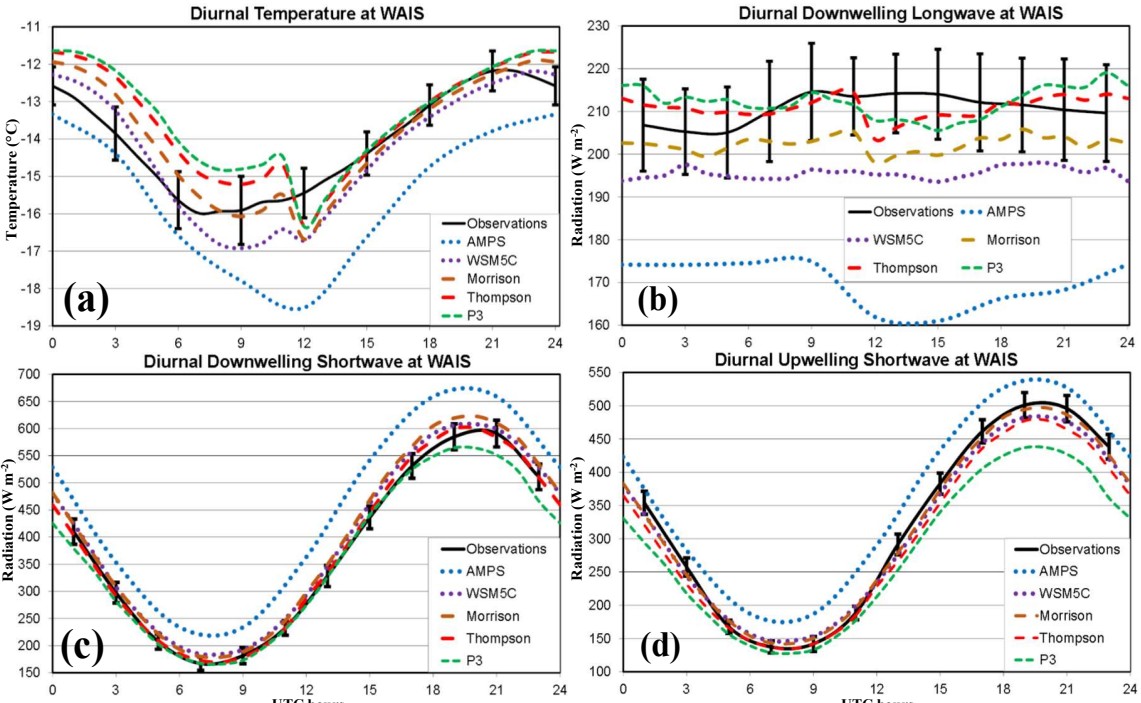

**Figure 5: Average diurnal cycles over 4 December 2015–19 January 2016 for (a) 2 m temperature (°C) and (b)-(d) over 7 December 2015–16 January 2016 for surface radiation (W m⁻²). (b) downwelling longwave radiation, (c) downwelling shortwave radiation, and (d) upwelling shortwave radiation. The error bars represent the 95% confidence level for differences between sample averages according the t-test (see text).**



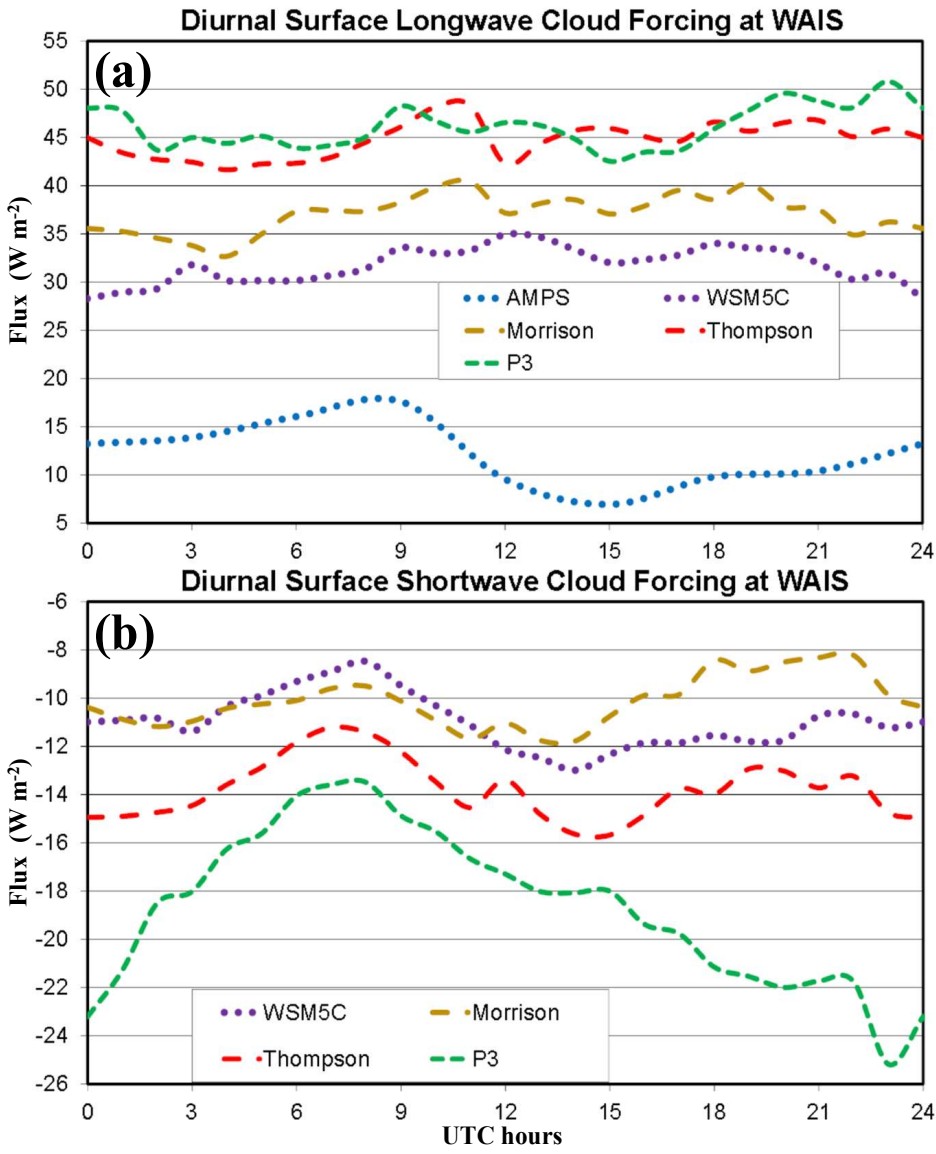

**Figure 6: Average diurnal cycles over 7 December 2015–16 January 2016 for (a) longwave cloud forcing (W m⁻²), and (b) shortwave cloud forcing (W m⁻²). AMPS values are shown in (a) as clear-sky values are available for longwave radiation, however, they are not available for shortwave radiation. Consequently shortwave cloud forcing was not calculated for AMPS.**



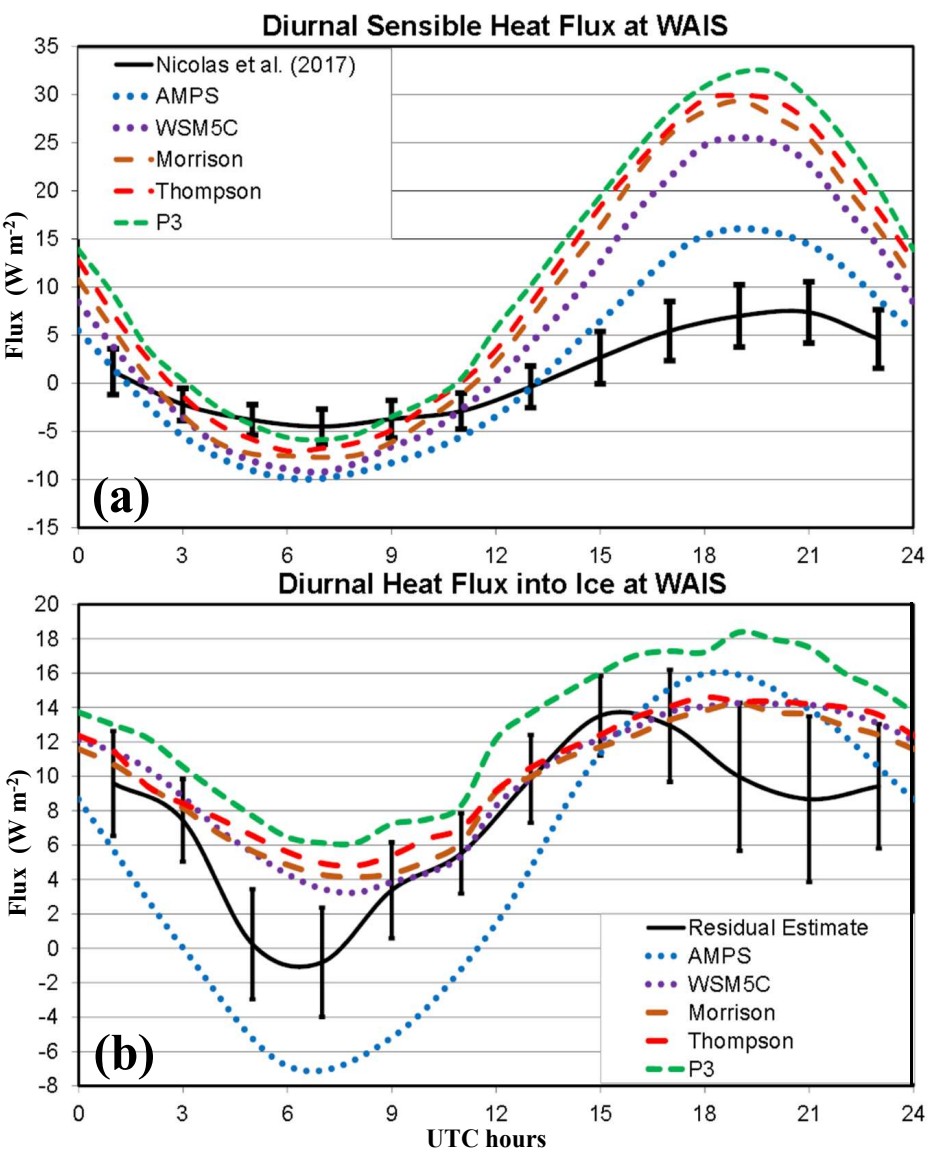

**Figure 7: Average diurnal cycles over 7 December 2015–16 January 2016 for (a) sensible heat flux (W m⁻²), and (b) heat flux into the ice pack (W m⁻²). WAIS observations are available for (a), while an estimate of the heat flux for (b) is available from the residual of surface energy balance terms.**





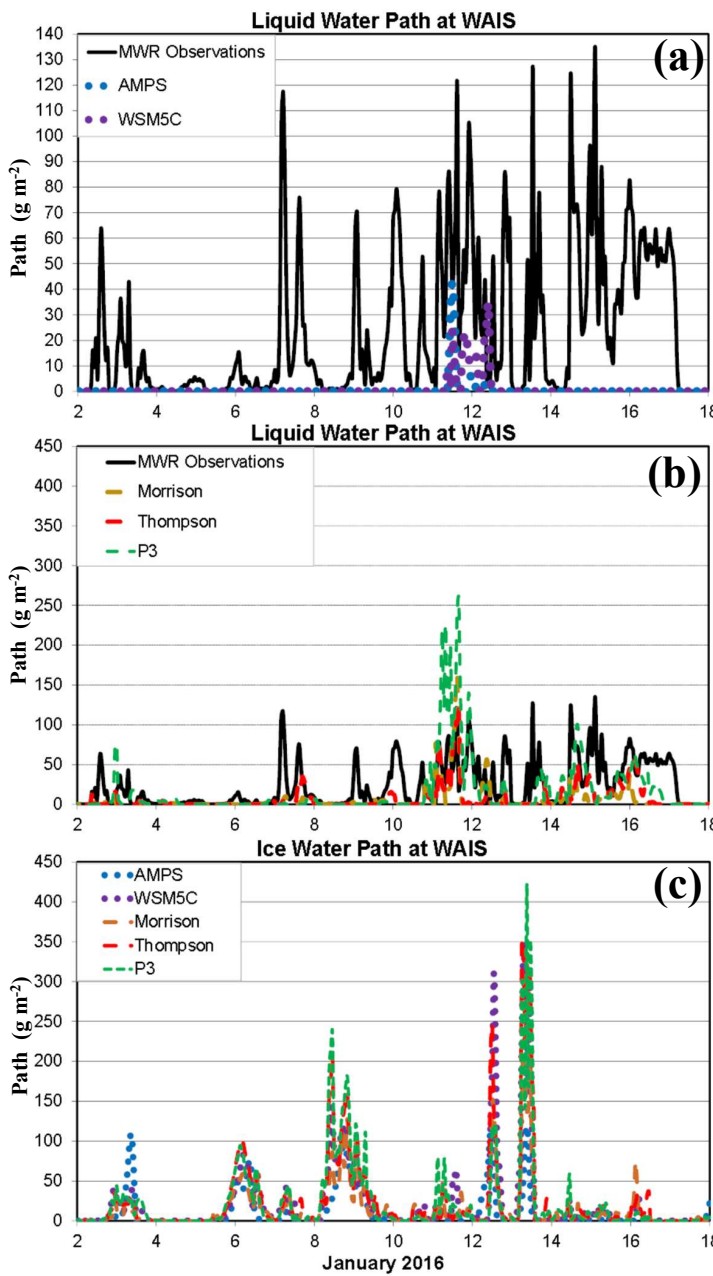

Figure 8: Time series of (a) and (b) liquid water path (mm) and (c) ice water path (mm) over 0000 UTC 2 January–0000 UTC 18 January 2016. Microwave radiometer (MWR) observations are available for liquid water path and are shown by solid curves in (a) and (b). Values for AMPS and the WSM5C simulation are shown in (a) and (c), while values for the three simulations with advanced microphysics schemes are shown in (b) and (c).

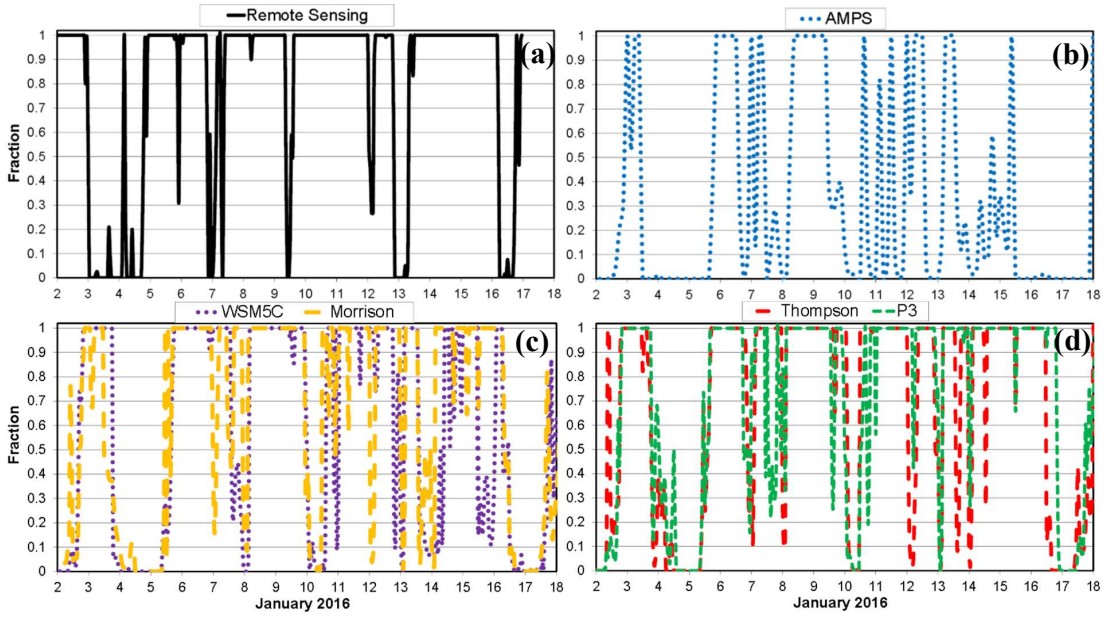

**Figure 9: Time series of cloud fraction for (a) remote sensing observations, (b) AMPS, (c) the WSM5C and Morrison simulations, and (d) the Thompson and P3 simulations. Model values of cloud fraction are based upon the Fogt and Bromwich (2008) algorithm using liquid water path and ice water path.**



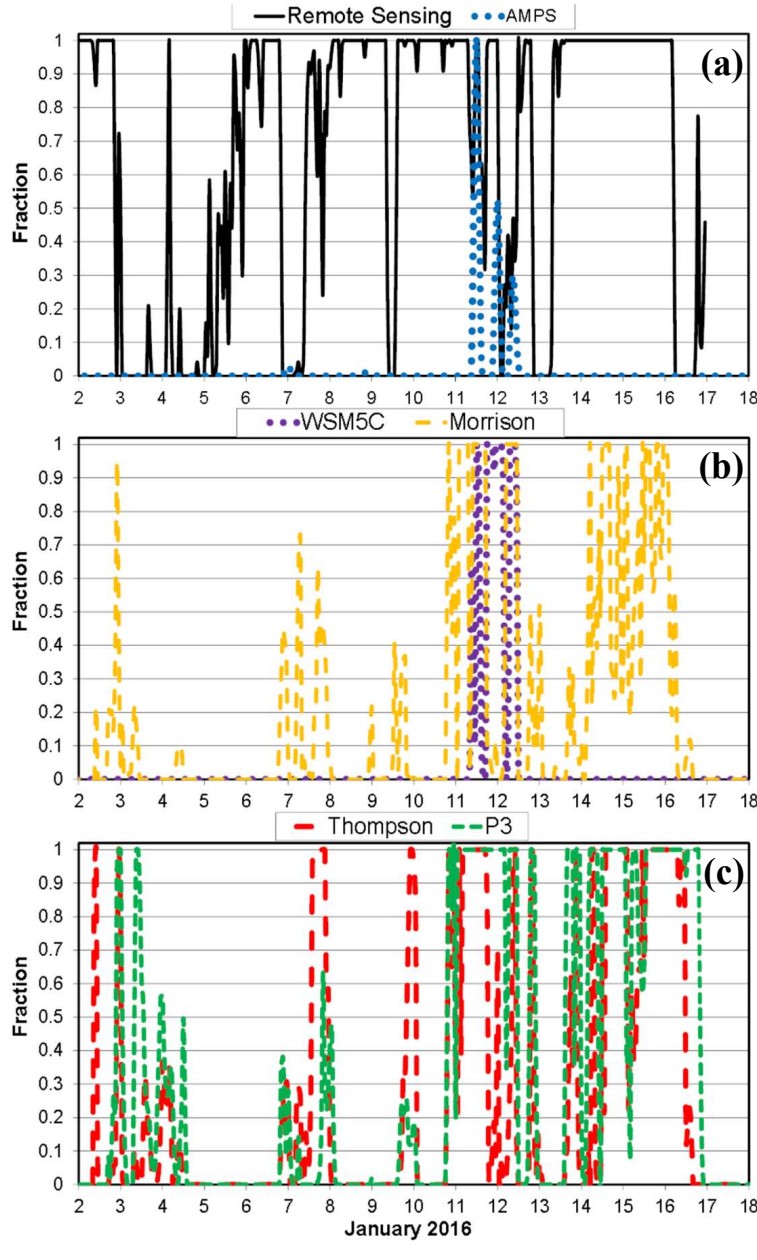

**Figure 10: Time series of liquid cloud fraction for (a) remote sensing observations and AMPS, (b) the WSM5C and Morrison simulations, and (c) the Thompson and P3 simulations. Model values of cloud fraction are based upon the Fogt and Bromwich (2008) algorithm.**





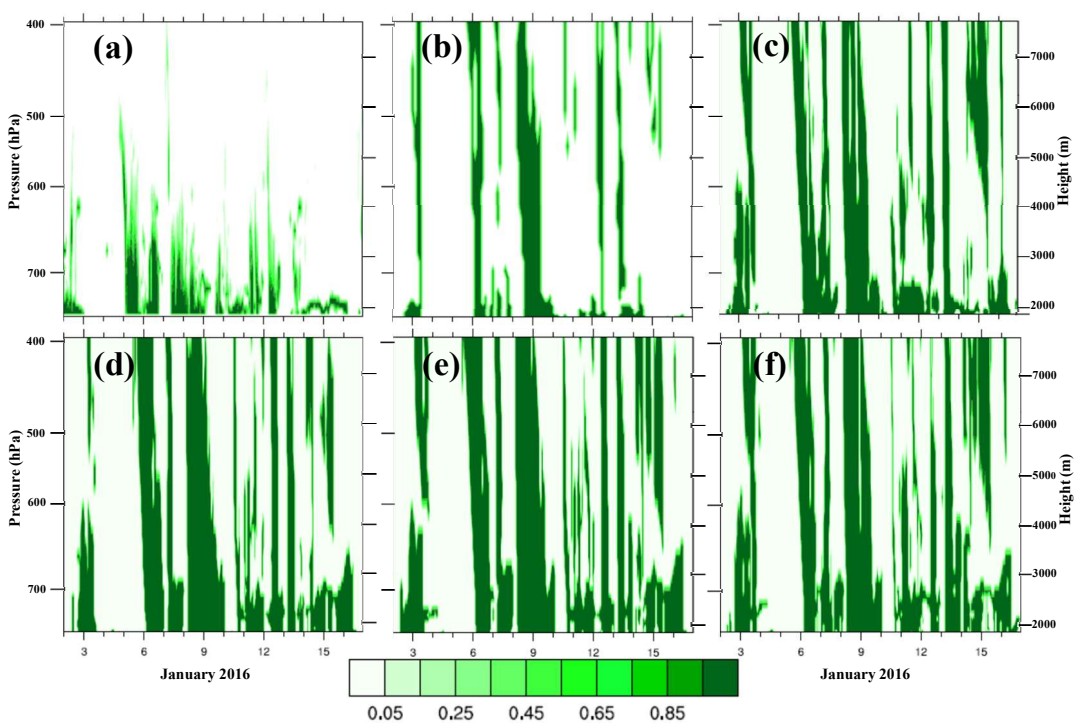

**Figure 11: Time-height plots of total cloud fraction (color scale) for (a) remote sensing observations, (b) AMPS, (c) WSM5C, (d) Morrison, (e) Thompson, and (f) P3. Model values of cloud fraction are based upon a condensate mixing ratio threshold of $10^{-6}$.**





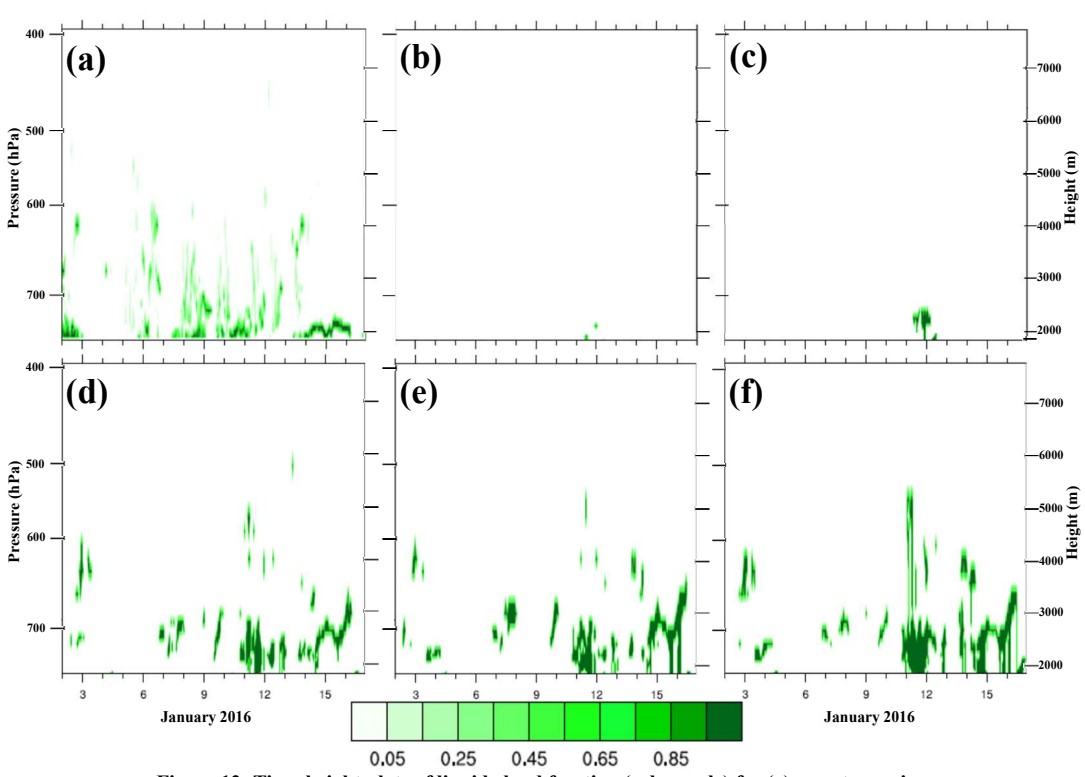

**Figure 12: Time-height plots of liquid cloud fraction (color scale) for (a) remote sensing observations, (b) AMPS, (c) WSM5C, (d) Morrison, (e) Thompson, and (f) P3. Model values of cloud fraction are based upon a condensate mixing ratio threshold of $10^{-6}$.**



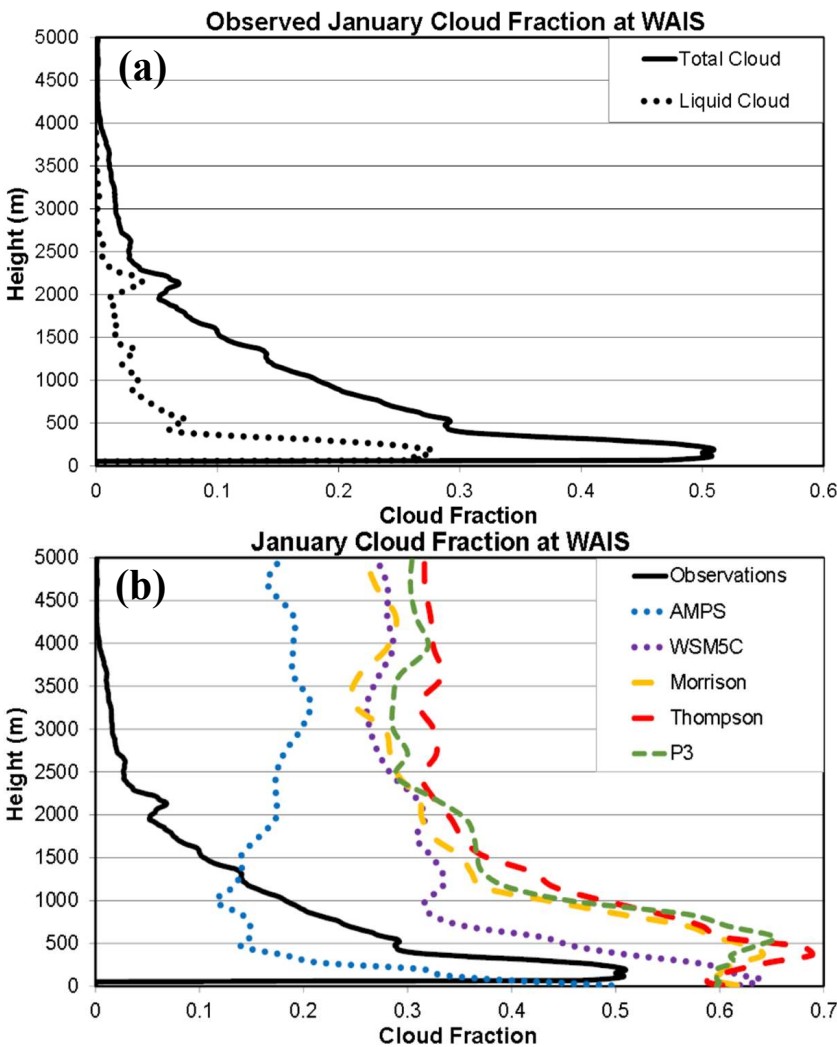

**Figure 13: Vertical profiles of average cloud fraction over 2–16 January
2016 for (a) remote sensing observations of total cloud fraction and liquid
cloud fraction, and (b) observations and simulations of total cloud fraction.**





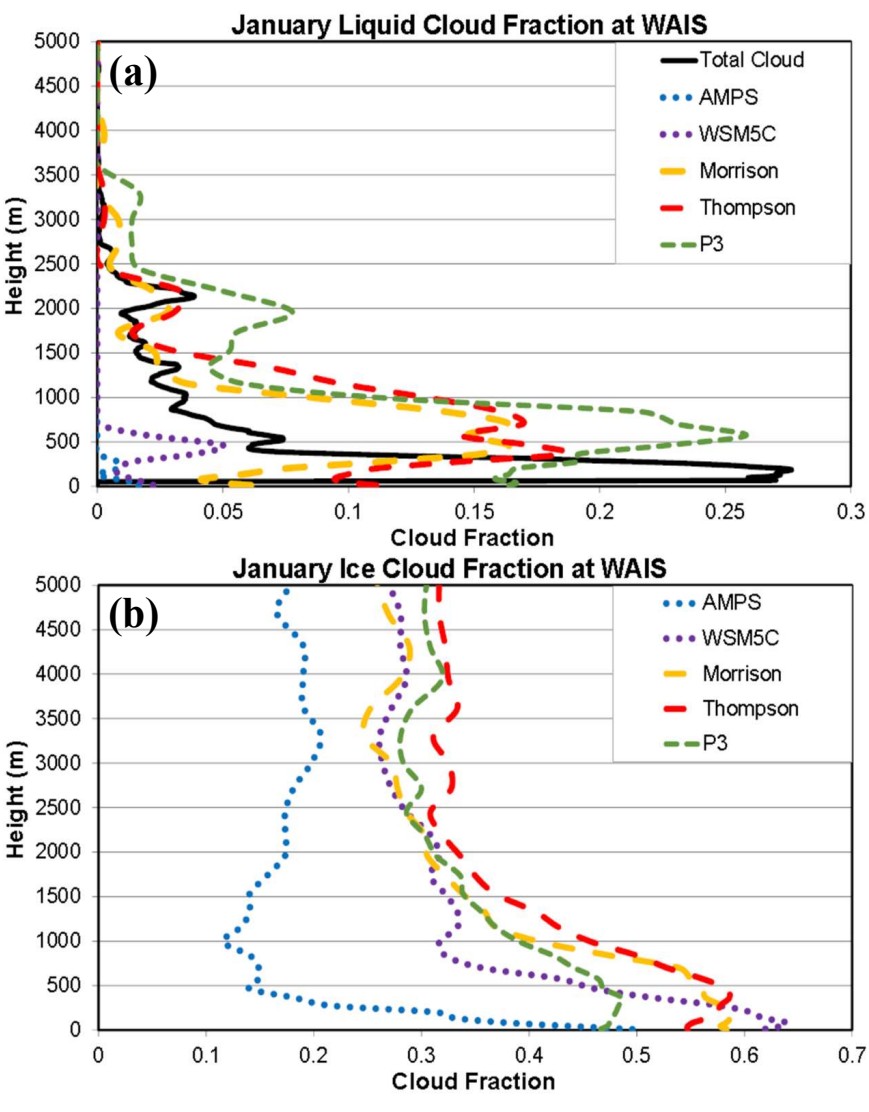

**Figure 14: Vertical profiles of average cloud fraction over 2–16
January 2016 for (a) liquid cloud fraction, and (b) ice cloud fraction.**





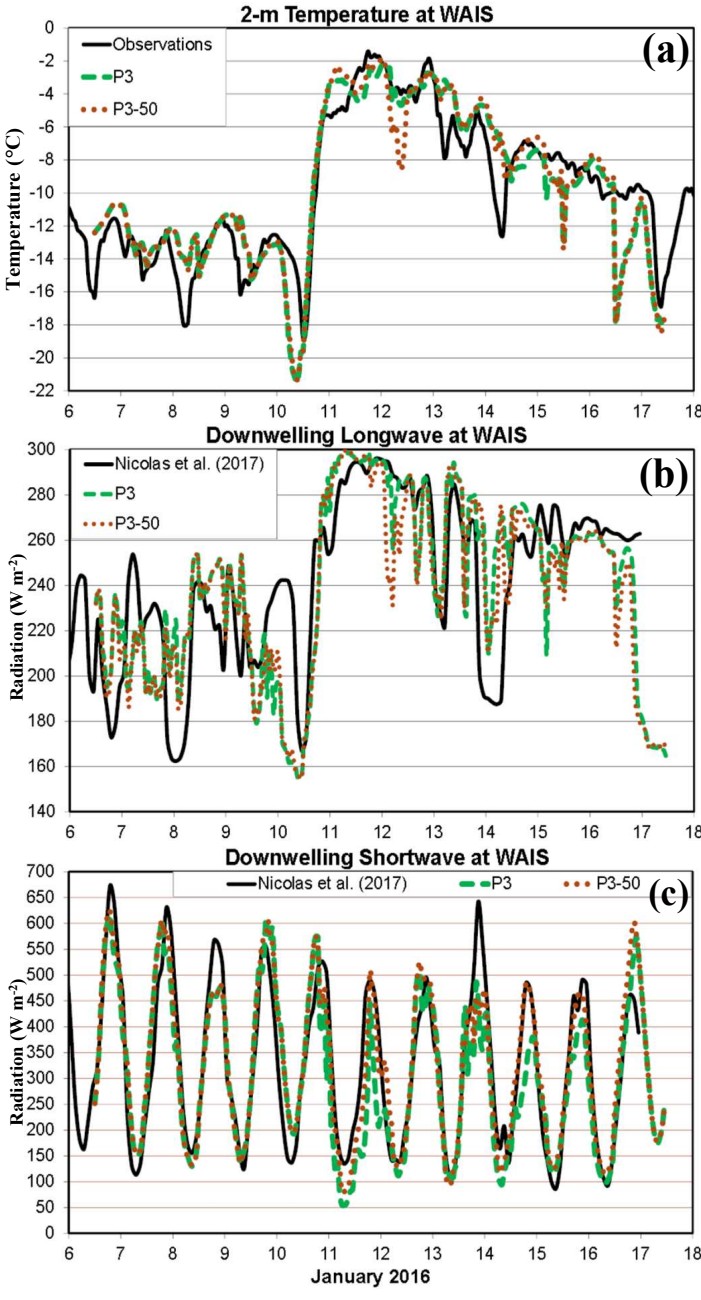

**Figure 15: Times series of (a) 2 m temperature (°C), (b) downwelling longwave radiation (W m⁻²), and (c) downwelling shortwave radiation (W m⁻²) during 6 January–17 January 2016 for the observations, the P3 simulation, and P3-50 sensitivity test.**