# Peer review of "Microphysics of Summer Clouds in Central West Antarctica Simulated by Polar WRF and AMPS"

_Atmospheric Chemistry and Physics, 2018_

## Referee Comment (RC1) · Anonymous Referee #1 · 29 Jan 2019

Summary

This is a study of Antarctic clouds, simulated in various versions of WRF, using the operational version (AMPS) with using GFS boundary conditions, and four versions of WRF v3.9.1 using ERA-Interim as boundary conditions, but with different moist physics schemes; some which are referred to as "more advanced". The model results are compared to observations from the ARM Mobile Facility deployment at the West-Antarctic Ice Sheet (WAIS) Divide. The authors suggest that biases in AMPS are related to a lack of clouds, and especially of liquid clouds, in AMPS and that the "more advanced" moist physics schemes to a large part alleviates this problem.

I must confess up front that I am quite skeptic to this type of study, where one plugs in different physics schemes without making sure that the whole model is well calibrated

[Figure]

– or tuned – for each scheme. We do not learn much from such an exercise beyond the pure technical lessons and in this case I remain unconvinced; my main impression is that the lions-hare of the improvements in fact comes from changing lateral boundary conditions, from GFS to ERA-Interim. Moreover, the manu-script is not particularly well organized or well written, and the methods used to evaluate the results are not particularly exiting. Therefore, I recommend that this manuscript is rejected at this time, but encourage the authors to come back with a better prepared and more solid study.

General comments

The authors motivate this study by the need to improve models, for operational fore-casting in Antarctica and for modeling the mass and energy budgets of the West-Antarctic Ice Sheet (WAIS), and putting the unique data from the AMF deployment AWARE to use, here especially at the WAIS Divide. However, these arguments pop up here and there in small pieces and it is unclear what the main underlying motivation is. For example, as far into the manuscript as in Section 3, we are told that the "primary concern" are cloud forecasts in AMPS. It shouldn't be too difficult to in a few sentences describing the background motivation and the particulars and leave it at that, without having to come back to the arguments again and again. There are also other parts of the paper that gives the impression that all the thoughts and ideas here were collected in a pile on the table and were not put together in a concise fashion.

The totality of all physics schemes in a model is a very complicated issue likely contain-ing a multitude of compensating errors, and for an optimal model all schemes need to be tuned to each other. Replacing one scheme with a completely different one without a proper retuning will of course produce a difference, but this may be due to the cre-ation of new differently compensating errors, or even improving results for reasons that may remain obscure. Hence, any conclusions that one scheme is better than another will be very difficult to draw from a study like this, and what might be a beneficial result might emanate from somewhere else in the model.

In the present case I'm actually unconvinced that the more modern schemes in fact does so much better than the default scheme in AMPS. AMPS is clearly an outlier in of these results, but replacing GFS with ERA-Interim and imposing the latter for a much smaller domain, as in WSM5C-run, the results become quite close to the other WRF v3.9.1 runs in everything but possibly the distribution between cloud liquid and ice. For all other parameters, I submit that the differences between the WRF v3.9.1 cloud have been removed if the different model version had been optimally tuned.

The authors make numerous statements about what biases are statistically significant or not but I wonder if the differences between the different versions of WRF v3.9.1 are in fact statistically significantly different from each other. Over-all, the error analysis is rather run-off-the-mill and unimaginative. Instead of endless tables with biases and correlation coefficients, I would have liked to see the full probability distributions of the errors and also some more imaginative error metrics. I would urge the authors to present a more process-related error analysis. It is not sufficient to plot different time series on top of each other, or even to plot time series of the errors. Instead, think about what processes are driving results for a certain variable, for example in an energy flux, and come up with ways to compare observations and the model for those parameters or sets of parameters. For example, the surface sensible heat flux is due to the surface and air temperatures (or their difference), the wind speed, and the eddy-exchange coefficient, all of which can be in error. In short, be a little imaginative. Below I will list numerous detailed comments that also need addressing.

Finally, the whole section on comparisons of cloud fractions, especially their profiles, should be revisited. This part of the study compares apples to pears; the model data should be run through a simulator that provides what the remote sensing instruments would have seen had the atmosphere looked like the model output. Time periods where the lidar was likely attenuated should be excluded completely. The model provides an instantaneous cloud fraction based on cloud condensate, while the remote sensing observations have to be averaged somehow over time. The model will give the same

cloud fraction for cases with the same LWP but a certain value of LWP can come from low cloud fraction and dense clouds as well as complete cover of less dense clouds.

Detailed comments

Page 2, line 19-28: This is a very broad background, even bringing in paleoclimatology. I wonder how the present study helps solve these issues? Is WRF used for climate scenarios and sea-level rise estimates?

P3, l5-6: Here argues that Antarctic clouds are different from Arctic, and low Antarctic IN concentrations is given as an example. But isn't that true also in the Arctic?

P3, l8: Arctic cloudiness peaks in boreal summer, but does cloud thickness? It seems to me that the thickest clouds would be frontal clouds and those do not dominate the Arctic summer cloudiness; the dominating clouds in the Arctic are low mixed-phase clouds that are not very thick. Maybe a ref-erence here would be appropriate.

P3, l10-11: The anthropogenic on aerosols in the Arctic is not particularly large in summer when clouds are at maximum, at least not over the Arctic Ocean.

P4, l3: I would have thought that improvement of atmospheric models was at the fore-front of the AWARE deployment, not ice-sheet models. Did the measurements include any ice-sheet relevant parameters?

P4, 6-9: This, at the end, seems like the relevant motivation here, so it should come first.

P4, l11: However valuable the AWARE deployment is, and it was indeed, it is not "climatological"; for that the AMF should have been left there at least 10 years; more actually.

P4, l12-14: How about marking the locations on the maps in Figure 1b.

P4, l6-7: This is probably similar to what the model is also doing, so it is not a mea-surement; its comparing one model to another. Was there no direct measurement of

**[ACPD](ACPD)**

Interactive
comment

sensible heat flux?

P4, l29-31: Was this done assuming there is no storage term? The surface energy does not need to be in balance; that term is misleading, and one should look at the energy budget, not the balance.

P5, l1: Do we need to know where this calculation was done?

P5, l1-5: Maybe some of this is unnecessary detail for this paper?

P5, l9-12: Here we need more detail, since this data is later used for observed cloud fraction pro-files, which are concluded to be significantly different from the model at higher altitude. What liquid water path does it take to extinguish the signal? Maybe time periods when LWP is larger than that should be excluded from the cloud analysis.

P6, l31: Maybe a reference for ASCOS?

P6, l4-5: Maybe "acute" is an overstatement?

P6, l11: 12 layers below 1km, is the kind of rather blunt information often given by modelers who do not want to disclose poor resolution close to the surface. Instead tell what the resolution is close to the surface and at what height the first model level is. That gives the reader a chance to determine if (s)he things the resolution is high enough.

P6, 19-22: Are these observations also assimilated in GFS? If so, are they effectively given double weight?

P6, l22: Reading this, what it says is that there are four forecast each day, two at 00UTC and two 12UTC.

P6, l25: Please use another word; fluctuations occur all the time, and here I assume you refer to steps that can occur every 12 hours going from one forecast to the next.

P6, l28-29: "Flights" of what?

P6, l1-2: If this is the primary goal, and I can believe it is, maybe this should have come in the introduction rather than improving ice-sheet modeling.

P6, l6-13: Here is central information about the model design that should have come up front, and not as an afterthought.

P6, l14: Using what at the lateral boundaries?

P6, l19: One golden rule in model testing is to change one thing at a time; here you use a different PBL scheme compared to AMPS. Why?

P6, l22: And what convection scheme is used in AMPS, not that I think convection is very important in Antarctica.

P6, l24-26: Is the ice fraction also different from AMPS?

P8, l14: And what is a Cooper curve? At least provide a reference.

P8, l18: As far as I understand, the ASCOS experiment was carried out in the Atlantic sector of the central Arctic; not in the eastern Arctic.

P8, l26: Awkward; cut this sentence after". . . can vary."

P8, l30-31: "water-friendly and ice-friendly" is hardly the appropriate terminology.

P8, l8 - P9, l6: In this section it would be useful to have a more in-depth discussion of what physics makes these schemes different, not only what variables they carry.

P9, l14-21: This is another example where central information is buried long into the text. First discuss all the aspects that is the same for the WRF v3.9.1 simulations and then discuss the differ-ences.

P9, l21: Is nudging done in AMPS?

P9, l23-26: Actually this far in I ask myself, why include the AMPS simulations at all? The first WRF v3.9.1 simulation is done as a baseline comparison to AMPS, with similar physics and the same moist physics scheme, but different lateral boundary forcing. But

there is an even larger difference here. Not only are different large scale fields used; the size of the "outer" domain is dramatically different. To really know what is what, there should have been an additional run exactly like WSM5C but with GFS imposed directly on smaller domain.

P10, l5: Actually not; there is no onset of anything demonstrated in this plot; just one static field. To show an onset requires a time line. Moreover, it is impossible to see the wind barbs in this plot.

P10, l11: The warming occurs "at" 12UTC; not after.

P10, l13: Pretty weak; "can be inferred". From what? Are there observations of some kind?

P10, l19-25: Actually the different microphysics tests are reasonably similar; the only big change is to AMPS.

P11, l1-6: There is a really strange behavior in the very lowest layer of the profiles, indicating a problem with the boundary conditions in the model. This raises the question of an analysis of e.g. 2-meter temperatures is at all meaningful. Also, here and elsewhere, there are problems with sign convention. I suggest assigning cold bias-es (e.g. line 6) negative values, not to confuse with positive biases (line 3).

P11, l19: Pretty bold statement. While I can agree that a bias on longwave radiation contributes to a bias in temperature, there are also other factors.

P11, l30-21: Sorry, but I don't understand this explanation.

P12, l14-15, There are differences, but to say that this the choice of microphysics "strongly impacts" temperature bias is an overstatement.

P12, l19-20 & l22-24: First the bias is not statistically significant and then "... all of these biases are statistically significant ...". Sound like a contradiction?

P13, l29 – P14, l14: First there is a discussion on satellite data, then a discussion

about the figure and then more discussions on satellite comparison (lines 8-14). Please organize the text better.

P14, l19-20: Is that small diurnal variation significantly different from a constant value?

P14, l26: Again, the community is moving away from the concept of a surface energy balance, because mostly there is no balance. It's a budget, and sometimes the sum of the fluxes are larger than zero and sometimes smaller. That is what makes the temperature change.

P15, l1: Is it a "bias" when you compare to a residual estimate? Maybe use "difference" instead.

P15, l19: Move "..., respectively" to the end of the sentence.

P16, l11: Fig. 9a?

P17, l3-4: This is not meaningful, since it is obvious the lidar do not capture any higher clouds; hence these are not "observed" by the observations.

P17, l21-22: This statement is largly unsupported.

P17, l24-33: This is just a long list of what the reader can see him/her self on the plots. What we need here is a synthesis.

P17, l29: What do you mean by "reflection"? There is no mirroring here what I can see.

---

## Referee Comment (RC2) · Anonymous Referee #2 · 14 Mar 2019

General comments

This paper compares radiation measurements with models runs with different microphysical schemes and meteorological fields. The paper has some interesting results and should eventually be suitable for publication, however I do have some concerns that should be addressed first.

I get the impression that the authors want to suggest that certain "advanced" microphysics schemes perform better than older schemes. However they need to be careful to discuss whether there data really support this. Have they fully explored other reasons for model measurement mismatch e.g. model boundary conditions, model physics, model resolution or measurement uncertainty. It seems clear that the differences between AMPS and WRF is predominantly down to the source of the meteoro-

logical data (e.g. GFS forecast data vs ERA-interim reanalysis).

The figures showing comparisons between model and measurement need to be significantly improved. Currently agreement cannot be properly assessed from the time series figures.

I really don't understand the section on cloud fractions. The authors calculate the cloud fraction as 0.075LWP + 0.170IWP, but the rest of the section seems to make out that this is a measure of cloud frequency of occurrence. Have I missed something? Even if I have this sections needs to be made clearer.

Specific comments

Page 2, Line 8 (and throughout) –Don't use the word "advanced". This is subjective and depends what you are comparing the scheme to. Try and keep the language as scientific as possible.

Page 3, Line 29 – I don't know what a "robust field program" is? Remove "robust".

Page 3, Line 30 to 10 (and throughout this section) – Most of this would be better in the introduction. This section should be a description of the field program and methods. The motivation for the project should come in the introduction.

Page 4, Line 10 – It is unclear to me what "well-calibrated" means? You don't explain how any of the instruments were calibrated. You should discuss this and also data uncertainty.

Page 4, Line 12- Suggest having a map with the field sites marked.

Page 5, Line 13 – You never explain what WRF stands for.

Page 7 line 18 – Please reword keeping the language as scientific as possible.

Page 8, Line 18 – O'Shea et al measurements were over the Weddell Sea not the Antarctic Peninsula.

Page 11, line 4 – Again stop referring to more "advanced" microphysics schemes, rather give the actual name(s) of the schemes.

Page 11, line 30 to 35 – I don't understand this? What is "observed difference was at the boundary of the 95% confidence level"?

Page 12, line 14 – Saying the microphysics scheme strongly impacts the temperature bias seems like an exaggeration.

Page 12, line 26 – Which schemes are you referring too when you say advanced? Which schemes are you comparing them with? If WSM5C is the less advanced scheme its performance looks comparable with Morrison and P3?

Page 13, line 12- Again this may not be due to the microphysics scheme. It could be related to the met fields, other characteristics of the model or uncertainty in the measurements.

Page 15, line 6 – The LWP retrieval and uncertainty should be discussed in the methods section (section 3).

Page 16, line 4 – I am not sure what the point of this metric is? You've already shown IWP and LWP plots, what is this metric and figures 9/10 really adding to the paper?

Page 16, line 16 – What does "Liquid cloud occurrence fraction" actually mean? Are you just multiplying the LWP by 0.075? This isn't a measure of the frequency that clouds occur. You either need to clarify or remove this analysis.

Page 19 Line 5 to 11 – This paragraph shouldn't be in the conclusions. A discussion of the aims of the project/work should come in the introduction.

Table 1 – Add source of meteorological fields. It would also be useful if you added other key characteristics of the schemes (IN parameterisation, number of habits, PSD parameterisation, etc)

Figure 8 to 12 – These times series plots are not clear, can't make out individual lines.

Suggest you consider other ways to show this data.

---

## Author Comment (AC1) · 8 May 2019

**Response to Interactive Reviewer 2's comments on "Microphysics of Summer Clouds in Central West Antarctica Simulated by Polar WRF and AMPS" by Hines et al.**

Response to General Comments

Thank you for these comments. We have changed the wording describing the microphysics schemes. We might add though that we believe our previous words are not "promotional" as much as they are accurate descriptions of the current thought in the cloud modelling community. The one-moment WSM5C scheme represents an older approach to cloud modeling with a prognostic treatment of several hydrometers in terms of the mixing ratio, while the other schemes are from more recent generations of cloud modelling and include elements of two-moment microphysics schemes. Thus, the newer schemes predict both the mixing ratio and some measure of the cloud size distribution. Accordingly, they allow for a greater degree of freedom in the cloud hydrometers. We have had extensive discussions with Hugh Morrison and Greg Thompson on these microphysics schemes.

Here is some text we included from the discussion with Reviewer 1:

We believe the comparison of the WSM5C microphysics schemes to the other schemes – which we refer to as more advanced schemes – is well founded. The WSM5C microphysics scheme is well-known in WRF modeling community to have difficulty simulating supercooled liquid water. More generally, representing supercooled liquid water is known to be difficult in numerical modeling studies. We have added the reference of Morrison and Pinto (2006) in this regard. Hugh Morrison's microphysics scheme, which was developed with the Arctic in mind is relatively successful in representing Arctic cloud water (Hines and Bromwich 2017 and references therein). This is known in the polar climate modeling community. So we believe the comparison of the WSM5C scheme – a one-moment microphysics scheme which is a relatively older generation algorithm – to newer generation schemes is a reasonable thing to do.

AMPS is considering changing microphysics schemes for better cloud representation. Other schemes, however, are more computationally expensive (Jordan Powers, personal communication, 2018), so the cpu cost must be weighed versus the gain in results. Our research is relevant to this decision.
* * *
Since the role of the different boundary conditions and initial conditions has been discussed by both reviewers, we added analysis of a simulation with WSM5C microphysics scheme that has forcing by the GFS final analysis rather than ERA-Interim. The new simulation is called WRF GFS. The results of this simulation show a 2 m temperature cold bias (-1.5 °C) similar to that of AMPS (-1.6 °C). The longwave and shortwave radiation for WRF GFS have biases of the same sign and slightly larger magnitude than those of WSM5C simulation. For longwave, the bias for WSM5C is -14.8 W m$^{-2}$ and -17.0 W m$^{-2}$ for WRF GFS. The results of the new simulation supports the

conclusion that the WSM5C microphysics has biases in the representation of clouds leading to too much downwelling shortwave and too little downwelling longwave radiation at the surface of West Antarctica.

We have done what we could to improve the figures. We recognize that the reduction in size to the manuscript specifications in the initial submission reduces the visibility of figures. We have attached selected larger figures to this response. We will see to it that high quality figures are sent to the journal for final publication.

Observations of clouds by surface observers or by remote sensing techniques differentiate between "cloud fraction" and "cloud frequency". For comparison to the model, the distinction is not so important since model cloud fraction tends to be zero or one (either by the vertically-integrated hydrometer method taken from Fogt and Bromwich [2008] or by a local threshold value set at a hydrometer mixing ratio of $10^{-6}$). We have sought to make the manuscript more clear on this point.

Specific Comments:

**Page 2, Line 8, Page 3, Line 29, Page 4, Line 10, Page 5 Line 13, Page 7 Line 18, Page 8 Line 18, Page 11 Line 4, Page 11 Line 30-35, Page 12 Line 14 and Page 12 Line 26.** The text has been modified to address these comments.

**Page 3, Line 30, Page 15, Line 6 and Page 19, Line 5-11.** The text has been rearranged based upon these comments.

**Page 4, Line 12**. The field site locations are added to Figures 1b and 2b.

**Page 13, Line 12.** The new simulation WRF GFS helps here in that it shows the radiation biases do not greatly change between forcing by GFS or ERA-Interim. The results are consistent with simulations with the WSM5C scheme showing too little liquid water, too little downwelling longwave radiation and too much downwelling shortwave radiation. This scheme is rather well known among WRF users in the polar regions for simulating too little liquid water. We have added the reference to Morrison and Pinto (2006) about the known difficulty of models simulating liquid water in polar regions. Our simulations with the more recent microphysics schemes produce more liquid water and greater cloud radiative effect.

**Page 16, Line 4.** The observations that could be taken at WAIS Divide during December 2015 and January 2016 were limited, due to the remoteness of the location. The main observational location for AWARE was at the McMurdo station that is a major freight transit point in Antarctica. We must use what observations are available for WAIS. Fortunately, lidar observations were available at WAIS, and the observations can be processed into cloud fraction following Sibler et al. (2018). We are making use of the available observations for comparisons to the modeling results.

**Page 16, Line 16.** Observations of clouds by surface observers or by remote sensing

techniques differentiate between "cloud fraction" and "cloud frequency". For comparison to the model, the distinction is not so important since model cloud fraction tends to be zero or one (either by the vertically-integrated hydrometer method taken from Fogt and Bromwich [2008] or by a local threshold value set at a hydrometer mixing ratio of $10^{-6}$). We have sought to make the manuscript more clear on this point.

**Table 1.** We have added the driving source of the meteorological fields to Table 1. We believe the detailed description of the microphysics schemes is best shown in Section 3.2.

**Figures 8-12**. Unfortunately, the similarity of the time series makes it difficult to differentiate some of the lines. Larger size versions of Figures 8-10 (old ordering according to the previous submission) are attached here for better clarity. Figures 11 and 12 have been replaced due to the introduction of the lidar simulator.

[Figure]

**Figure 8: Time series of (a) and (b) liquid water path (mm) and (c) ice water path (mm) over 0000 UTC 2 January–0000 UTC 18 January 2016. Microwave radiometer (MWR) observations are available for liquid water path and are shown by solid curves in (a) and (b). Values for AMPS and the WSM5C simulation are shown in (a) and (c), while values for the three simulations with advanced microphysics schemes are shown in (b) and (c).**

[Figure]

Figure 9: Time series of cloud fraction for (a) remote sensing observations, (b) AMPS, (c) the WSM5C and Morrison simulations, and (d) the Thompson and P3 simulations. Model values of cloud fraction are based upon the Fogt and Bromwich (2008) algorithm using liquid water path and ice water path.

[Figure]

**Figure 10: Time series of liquid cloud fraction for (a) remote sensing observations and AMPS, (b) the WSM5C and Morrison simulations, and (c) the Thompson and P3 simulations. Model values of cloud fraction are based upon the Fogt and Bromwich (2008) algorithm.**

---

## Author Comment (AC2) · 8 May 2019

**Response to Interactive Reviewer 1's comments on "Microphysics of Summer Clouds in Central West Antarctica Simulated by Polar WRF and AMPS" by Hines et al.**

Response to summary.

Perhaps it is important here to emphasize our motivation. The AMPS forecasts are widely used in Antarctica and support operations – including aircraft flights – in this difficult and extreme environment (Bromwich et al. 2005; Powers et al. 2009, 2012, Wille et al. 2017). The weakness in representing clouds has been known for some time. A former member of the Polar Meteorology Group at The Ohio State University did a Master's Thesis that looked at the representation of clouds in AMPS (Pon 2015). Also, we have plenty of experience running Polar WRF in both hemispheres (e.g., Bromwich et al. 2013, 2018) and this includes looking at the representations of clouds by Polar WRF in the Arctic (Hines and Bromwich 2017). We are highly motivated to study how well AMPS is doing in representing Antarctic clouds and how such forecasts might be improved. The recent AWARE project (2015-2017) was an obvious opportunity enabling working with AMPS cloud issues.

We took care to avoid overarching statements about how one of the newer microphysics schemes was generally better than the others, since extensive testing would be required make such general statements. The observations at WAIS Divide during December 2015-January 2016 are not detailed enough to show comprehensive ice and liquid cloud microphysics. In particular there is little direct measurement of cloud ice beyond generic "cloud". More extensive measurements are available at McMurdo. That site, however, is strongly influence by the detailed topography of Ross Island, while WAIS Divide has greater regional representativeness. We prefer to start with WAIS Divide for this reason. Additional work will be done with the more detailed measurements at McMurdo, but we believe we should be familiar with the characteristics of WAIS Divide first.

The existing combination of cloud and microphysics observations at WAIS Divide, nevertheless, enable many comparisons of model to observations. Model biases in cloud water, for example, can be expected to be revealed. Our results do show more liquid simulated with some schemes, especially those that include elements of two-moment microphysics. The expected impact of liquid water on radiation is demonstrated in the simulation results.

We believe the comparison of the WSM5C microphysics schemes to the other schemes – which we refer to as more advanced schemes – is well founded. The WSM5C microphysics scheme is well-known in WRF modeling community to have difficulty simulating supercooled liquid water. More generally, representing supercooled liquid water is known to be difficult in numerical modeling studies. We have added the reference of Morrison and Pinto (2006) in this regard. Hugh Morrison's microphysics scheme, which was developed with the Arctic in mind is relatively successful in representing Arctic cloud water (Hines and Bromwich 2017 and references therein). This is known in the polar climate modeling community. So we believe the comparison of the

WSM5C scheme – a one-moment microphysics scheme which is a relatively older generation algorithm – to newer generation schemes is a reasonable thing to do.

AMPS is considering changing microphysics schemes for better cloud representation. Other schemes, however, are more computationally expensive (Jordan Powers, personal communication, 2018), so the cpu cost must be weighed versus the gain in results. Our research is relevant to this decision.

We added some scatter plots for a different method of model vs. observation analysis than shown in the original submission of the manuscript. The new figure is shown here. In Fig. 4a, the negative temperature bias in AMPS is shown to be larger when the observed temperature is above about -10ºC. Thus, AMPS is unlikely to well represent melting events. The error in longwave radiation shown in Fig. 4c is larger when the observed longwave radiation larger than about 200 W m$^{-2}$. That is AMPS is less accurate at times when clouds are likely to be present. In contrast, Morrison, Thompson and P3 better treat cases when the observed longwave radiation is relatively large. The AMPS error tends to be smaller with the longwave radiation is relatively small. That is, the error tends to be smaller when cloudiness is small.

We have also added the lidar simulator from the CR-SIM Cloud Resolving Model (CRM) Radar Simulator version 3.2 for better comparison between modelled hydrometers and remote sensing of the clouds at WAIS Divide. This simulator has been used with WRF results.

[Figure]

**Figure 4:** Scatter plots of observed values (horizontal axis) and simulated results (vertical axis) of 2 m temperature (°C) for (a) AMPS, WSM5C and WRF GFS and (b) Morrison, and downwelling longwave radiation (W m$^{-2}$) for ( c) AMPS, WSM5C, and WRF GFS and (d) Morrison, Thompson and P3. The dashed line shows the 1 to 1 line.

Detailed Comments:

**Page 2 line 19-28, P3 l5-6, P3 l10-11, P4 l3, P4 l11, P 5 l1, P5 l1-5, P 6 l4-5, P6 l22, P6 l25, P6 l28-29, P8 l26, P10 l11, P10 l13, P12 l14-15, P14 l19-20, P15 l1, P16 l11 and P17 l29.** The text has been modified to address these comments.

**P3. l8.** We have gone back and checked the Sibler et al. (2018) reference, and the modified text is consistent with the reference.

**P4 l6-7.** We were unable to connect this comment to any line in the first version of the manuscript.

**P4 6-9.** The text has been rearranged based upon this comment.

**P4, l12-14**. The AWARE site locations are added to Figure 1b and 2b.

**P 4 l l29-31 and P14 l26**. Thank you for proving the most recent community viewpoint on how the surface thermodynamic equation should be treated. First, we should provide some background on our use of the "surface energy balance" for Table 3 and Figure 7b. We had hoped to use the measurements of the conductive flux in the ice pack at WAIS Divide. Unfortunately, instrument errors resulted measurements of unacceptable quality. Previously, Nicolas et al. (2017) produced alternative estimates of the conductive flux by assuming a balance of terms, then solving for the "ground" term. This work was presented in the work published in refereed journal *Nature Communications*. The storage term could, of course, be large instantaneously, but should have a relatively small value when averaged over time compared to other terms in the thermodynamic equation. We think then this method provides a reasonable estimate of the conductive flux, given that quality direct measurements were unavailable. Again, these are previously published numbers.

**P5 l9-12.** To compensate for the attenuation of the lidar signal by hydrometers, we have added the lidar simulator from the CR-SIM Cloud Resolving Model (CRM) Radar Simulator version 3.2 to comparison between model-simulated hydrometers and remote sensing observations at WAIS Divide. This simulator is configured for WRF model output.

**P5 l31**. A reference is added for Tjernström et al. (2014).

**P6 l1-2 and P6 l6-13.** The text has been rearranged based upon these comments.

**Page 6 l11**. We added information on the levels. The lowest levels are at 10, 37, 73, and 119 m.

**P6 l14**. The smaller domains shown in Fig. 1a are nested domains. Thus, they are "forced" by the larger domains. We have modified the text slightly.

**P6 l19.** It was not possible to equalize all settings between AMPS and the Polar WRF 3.9.1 simulations. This was an important reason for the inclusion of the WSM5C simulation, since it would have the same microphysics scheme as AMPS, yet have the same settings, except for the microphysics scheme, as the Morrison, Thompson, and P3 simulations. Since we ultimately wished to compare our results to the observations at WAIS Divide, it was desirable to have a good framework for our comparison. We used the PBL scheme that we thought would give the best results. The addition of new

simulation WRF GFS, discussed later, helps to bridge the gap between AMPS and WSM5C.

**P6 l19-22**. Large-scale data assimilation seeks to include many observations to set the analysis field. This may result in a smoothing of fields. Mesoscale data assimilation seeks to include mesoscale structures in the resulting field. So the goals of global data assimilation and mesoscale data assimilation are different. The risk/reward calculations are different. Mesoscale data assimilation tends to be more dependent on individual observations, as the goal is to represent fine features. An individual observation can influence both the global analysis field and the mesoscale data field derived in part from the global analysis field. In that sense the observation is "double dipping" but this is not an error. The key here is that data assimilation on different scales has different goals.

**P6 l24-26**. The AMPS source for sea ice fraction is now shown in the revised manuscript.

**P8 l14**. A reference is added for Cooper (1986).

**P8 l18**. Apparently, there is not a consensus as to the descriptions "Western Arctic" and "Eastern Arctic". We have changed the description of location of ASCOS in the revised manuscript.

**P8 l30-31**. The terminology "water friendly" and "ice friendly" is taken from the publication Thompson and Eidhammer (2014). We have had previous extensive discussions with Greg Thompson about this scheme. The wording has been changed in the revised manuscript.

**P8 l8- P9 l6**. We have added some words in section 3.2 on the differences between microphysics schemes.

**P9 l14-21.** The text has been rearranged based upon this comment.

**P9 l21**. We have checked and found no nudging is done in AMPS. There was some confusion in the preparation of the original manuscript because of a presentation by a former graduate student at Ohio State on the positive impact of "grid nudging" in WRF Antarctic forecasts that was inspired by AMPS forecasts. The manuscript has been changed to avoid confusion.

**P9 l23-26**. Please see the response to the summary explaining the importance of AMPS. We have added a simulation "WRF GFS" with the WSM5C microphysics and the GFS final analysis providing the initial and boundary conditions. We believe this helps to bridge the gap between the AMPS results, driven by the GFS forecast fields and the WSM5C simulation with Polar WRF 3.9.1 and driven by ERA-Interim.

**P10 l5**. It was not our intent to re-demonstrate in detail the West Antarctic warming discussed in the published paper Nicolas et al. (2017). The use of the word "demonstrated" in the original manuscript was unfortunate. The text has been changed.

We will take care in the submission of the final figures for quality and visibility. Unfortunately, the small size of figures in the first version limited visibility.

**P10 l19-25**. Yes, the difference between the simulations Morrison, Thompson and P3 is relatively small. No definitive claim can be made of superiority between these schemes. That could imply the fine detail differences between these more recent schemes has relatively small impact on the simulation results.

As to the WSM5C scheme, the addition of the new WRF GFS simulation helps. It has the same microphysics scheme as the WSM5C scheme, however it uses the GFS final analysis for initial and boundary conditions. This is not exactly the same as the GFS forecast used by AMPS (the final analysis is not available at forecast time, and AMPS is run prognostically). However, the GFS final analysis uses the same forecast system as the GFS forecasts. The simulations with the WSM5C scheme consistently produce too little cloud liquid, whether GFS or ERA-Interim is used for the initial and boundary conditions. Correspondingly, downwelling shortwave simulation is excessive and there is a deficit in downwelling longwave radiation. This is consistent with the experience of mesoscale modellers in the Arctic. The reference, Morrison and Pinto (2006), now used in the revised version, mentions that simulating supercooled liquid water is a known difficulty in the polar regions. Hugh Morrison's double-moment scheme has been known simulate supercooled water relatively well in multiple Arctic studies (several references are given in our earlier paper Hines and Bromwich 2017). In the present work, the three more recent microphysics schemes produce more liquid water, and have greater cloud forcing.

**P11 1-6**. The 2-m air temperature is close to the skin temperature, and the skin temperature is used for upwards longwave radiation at the surface and the calculation of conductive flux in the snowpack by the WRF Noah land surface model. So this temperature is important for the surface energy terms and the interaction therein. Therefore we choose to show the 2-m temperature in this paper. The 2-m temperature is also a widely-measured quantity, and at the height or near the height at which many other near-surface variables are measured.

Now, the surface boundary layer is of interest for the AWARE project, but our interest in this paper is the clouds and the related radiation. So we prefer not to divert attention away from the clouds and radiation by additional analysis of the boundary layer in this paper. We may look in greater detail at the boundary layer in our near-future AWARE work. This will probably involve the McMurdo observations that are more detailed than the WAIS divide observations.

The words have been changed about the description of biases in response to this comment. "Negative bias" and "positive bias" are now used in the text, and "cold bias" is less used in the revised manuscript.

**P11 l19.** The sentence is removed.

**P11 l30-21**. We change the explanation of how we determine the statistical significance.

**P12 19-20 & l22-24**. Perhaps it's understandable how discussion of the statistical significance of results for specific hours of the day versus that for all times could be confusing. We now mention in the text that Table 3 shows the biases for all times (rather than the bias for a specific time of day). It is easier to meet the criteria for statistical significance for all times, rather than for a specific time of day when the sample size is reduced.

**P13 l29- P14 l14**. We rearranged the discussion of the earlier satellite data studies in response to this comment.

**P17 l3-4.** We added the lidar simulator from the CR-SIM Cloud Resolving Model (CRM) Radar Simulator version 3.2 to comparison between model-simulated hydrometers and remote sensing observations at WAIS Divide. This simulator is configured for WRF model output.

**P17 l21-22.** We removed the sentence.

**P17 l24-33.** We removed some of the previous text. We added the sentence, "Similar to the profile displayed in Fig. 13a, the observations show a more shallow peak in the lower troposphere than in the simulations. (Fig. 14a)." The figure numbers in this reply are based upon the original submission of the manuscript.